# TRACING REPRESENTATION PROGRESSION: ANALYZING AND ENHANCING LAYER-WISE SIMILARITY

**Jiachen Jiang, Jinxin Zhou & Zhihui Zhu**
Department of Computer Science and Engineering,
The Ohio State University,
{jiang.2880, zhou.3820, zhu.3440}@osu.edu

## ABSTRACT

Analyzing the similarity of internal representations within and across different models has been an important technique for understanding the behavior of deep neural networks. Most existing methods for analyzing the similarity between representations of high dimensions, such as those based on Centered Kernel Alignment (CKA), rely on statistical properties of the representations for a set of data points. In this paper, we focus on transformer models and study the similarity of representations between the hidden layers of individual transformers. In this context, we show that a simple sample-wise cosine similarity metric is capable of capturing the similarity and aligns with the complicated CKA. Our experimental results on common transformers reveal that representations across layers are positively correlated, with similarity increasing when layers get closer. We provide a theoretical justification for this phenomenon under the geodesic curve assumption for the learned transformer, a property that may approximately hold for residual networks. We then show that an increase in representation similarity implies an increase in predicted probability when directly applying the last-layer classifier to any hidden layer representation. This offers a justification for *saturation events*, where the model's top prediction remains unchanged across subsequent layers, indicating that the shallow layer has already learned the necessary knowledge. We then propose an aligned training method to improve the effectiveness of shallow layer by enhancing the similarity between internal representations, with trained models that enjoy the following properties: (1) more early saturation events, (2) layer-wise accuracies monotonically increase and reveal the minimal depth needed for the given task, (3) when served as multi-exit models, they achieve on-par performance with standard multi-exit architectures which consist of additional classifiers designed for early exiting in shallow layers. To our knowledge, our work is the first to show that one common classifier is sufficient for multi-exit models. We conduct experiments on both vision and NLP tasks to demonstrate the performance of the proposed aligned training.

## 1 INTRODUCTION

As one of the most significant breakthroughs in deep neural network (DNN) architectures developed in recent years, the transformer model (Vaswani et al., 2017) has driven recent advances in various vision and NLP tasks, such as vision transformer for image classification (Dosovitskiy et al., 2020) and image generation (Yu et al., 2021; Ramesh et al., 2021; Yu et al., 2022), BERT (Devlin, 2018), GPT (Radford et al., 2019), and other various large language models (LLMs) (Zhao et al., 2023) for natural language understanding and generation. It has been viewed as a promising foundation model that can be adapted and extended to various applications and domains (Bommasani et al., 2021). Additionally, researchers have found that increasing the size (by stacking more layers and/or making them wider) can consistently improve performance, resulting in models of significant size (e.g., the 175B-parameter GPT-3 and the 540B-parameter PaLM). However, the ever-increasing size has posed a significant challenge in studying and understanding exactly how these models solve tasks and in efficiently deploying them.

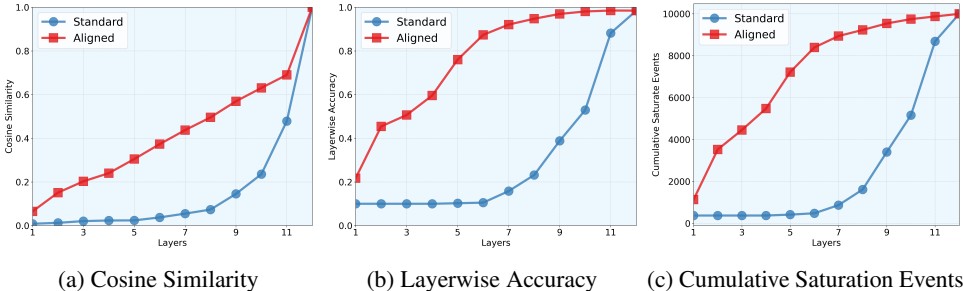

| (a) Cosine Similarity | (b) Layerwise Accuracy | (c) Cumulative Saturation Events |

Figure 1: Illustration of the DeiT-S (Touvron et al., 2021) (pretrained on ImageNet) fine-tuned on CIFAR-10 with standard method and the proposed aligned training in terms of (a) cosine similarity of features from shallow and the last-hidden layer, (b) layer-wise testing accuracies by applying the last-layer classifier to each layer, as well as (c) cumulative saturation events (Geva et al., 2022). We observe that our proposed aligned training can substantially enhance layer-wise representation similarity, thereby improving layer-wise accuracies and promoting more early saturate events.

Given the success of deep learning models, attributed to their ability to learn increasingly complex internal representations as they go deeper through their layers, a promising direction for understanding these models is to study the hierarchical feature learning across layers. Recent work (Papyan et al., 2020; Fang et al., 2021; Zhu et al., 2021; Thrampoulidis et al., 2022; Tirer et al., 2023) have uncovered an intriguing phenomenon regarding the last-layer features and classifier of DNNs, called Neural Collapse ($\mathcal{NC}$), across many different datasets and model architectures. Roughly speaking, $\mathcal{NC}$ refers to a training phenomenon in which the last-layer features from the same class become nearly identical, while those from different classes become maximally linearly separable. Beyond the last-layer features, recent studies have also shown that deep classifiers progressively compress within-class features while enhancing the discrimination of between-class features from shallow to deep layers (He & Su, 2023; Rangamani et al., 2023; Wang et al., 2023). Another line of work attempts to compare the representations within and across DNNs. Various approaches have been proposed to quantify the representation similarity, such as the Canonical Correlation Analysis (Thompson, 2000), Centered Kernel Alignment (CKA) (Kornblith et al., 2019), Orthogonal Procrustes Transformation (Hamilton et al., 2016) and Pointwise Normalized Kernel Alignment (PNKA) (Kolling et al., 2023). Representation similarity analysis has also been widely used in computational psychology and neuroscience as well (Edelman, 1998; Kriegeskorte et al., 2008).

As these approaches are designed to be invariant to certain transformations (such as orthogonal transformation) and can be applied to features with possibly different dimensions, they rely on statistical properties of the representations for entire training data. For instance, the $\mathcal{NC}$ analysis captures the variance of the features from each class, while the widely used CKA for representation analysis replies on the inter-example structures; see Section 2 for the detailed definition of CKA. Consequently, it has been observed that CKA is sensitive to outliers and may give unexpected or counter-intuitive results in certain situations (Davari et al., 2022). In this paper, we are motivated by the following question: *How are the representations of individual inputs progressively transformed from shallow to deep layers?*

**Contribution** In this work, we focus on transformer models, which have particular properties compared to other architectures: a transformer contains identical blocks with residual connections in each block. Thus, the features in a transformer model have the same dimension and may exhibit less rotation difference across layers. Motivated by this observation, we study the layer-wise representation similarity for transformer models on a per-sample basis, allowing us to directly apply the last-layer classifier right after any hidden layer for classification or text generation tasks. This enables an effortless multi-exit model that allows early exit during inference, thereby saving computation time. Our contributions can be summarized as follows.

- **Sample-wise cosine similarity captures representational similarity** We introduce a straightforward yet efficient sample-wise cosine similarity metric to examine the similarity of internal representations in transformers. Experiments show that the cosine similarity aligns with CKA, which is based on statistical properties of all the features, and is sufficient to reflect representation similarity. In addition, as illustrated in Figure 1 (a), our experimental results (Standard) on common transformers show that representation similarity increases as layers become closer. We

provide a theoretical justification for this phenomenon under the geodesic curve assumption for the learned transformer, a property that approximately holds for residual networks (Gai & Zhang, 2021), and hence for transformers.

- **Analysis for saturation events** The similarity with last-layer representation suggests that the last-layer classifier can be directly applied right after any hidden layers for decision-making, also known as the logit lens approach. Geva et al. (2022) discover a phenomenon called *saturation events*, where the model's final prediction becomes the top candidate in a certain shallow layer and remains unchanged across all subsequent layers, indicating that the shallow layer has already learned the necessary knowledge. We show that an increase in representation similarity implies an increase in predicted probability across layers. This offers a justification for saturation events, stating that if a sample is correctly predicted at the $\ell$-th layer, it will continue to be correctly predicted in subsequent layers, as the predicted probability increases progressively across layers.

- **Aligned training for enhancing layer-wise representation similarity** We propose an aligned training method to improve the effectiveness of shallow layers by increasing the similarity of internal representations between different layers. Motivated by the $\mathcal{NC}$ phenomenon, where features from the last-hidden layers align with the common classifier, our aligned training approach deploys the common classifier to each layer and then minimizes the average of the cross-entropy losses from all the layers. As shown in Figure 1, the aligned training can substantially enhance layer-wise representation similarity, thereby leading to more early saturation events and improving layer-wise accuracies. Consequently, the aligned training method can help identify the minimal number of layers needed by unleashing the power of shallow layers to transform features faster towards classifier across layers and push the redundancy behind.

- **Multi-exit models with a single classifier** Another important application of the proposed aligned training is to improve the inference efficiency of large models. During inference, the model allows early exit to save computation time. Previous works (Xin et al., 2020; Geng et al., 2021; Xin et al., 2021) design multi-exit models by introducing different classifiers to each layer, which may substantially increase the model size for a large number of classes, such as ImageNet with 1000 classes and GPT3 (Brown et al., 2020) with a vocabulary of $50,257$ tokens. Instead of using separate classifiers for each exit, our multi-exit model employs a common classifier, *which to our knowledge is the first of its kind*, maintaining the early exit capability and achieving performance on par with models that use multiple classifiers. We demonstrate the performance of the proposed aligned training method in both pretraining ViT and fine-tuning LLMs in NLP tasks, including fine-tuning BERT (Devlin, 2018) for text classification tasks on the General Language Understanding Evaluation (GLUE) benchmark (Wang et al., 2018) , and GPT2 (Radford et al., 2019) model for text generation on the Wikitext-103 dataset (Merity et al., 2016).

The MatFormer (Kudugunta et al., 2023) introduces a nested structure into the Transformer by jointly training all submodels of different widths. In contrast, our method jointly trains submodels with varying layers. Exploring the potential integration of these approaches will be a focus of future research. While we mainly focus on transformer models, the proposed aligned training can be applied to other deep architectures, provided they have the same dimensions in each layer. Extending this approach to accommodate varied feature dimensions is the subject of future work.

## 2 Measuring Layer-wise Representational Similarity

In a transformer $f$ with $L$ layers, the representations gradually evolve across layers, with the progression from one layer (say $\ell$-th layer) to the next following a residual update pattern:

$$\boldsymbol{H}^{(\ell+1)} = \boldsymbol{H}^{(\ell)} + f_{\theta^{(\ell)}}(\boldsymbol{H}^{(\ell)}), \tag{1}$$

where $\boldsymbol{H}^{(\ell)} \in \mathbb{R}^{d \times s}$ are input feature sequence of length $s$ with hidden dimension of $d$. The residual block $f_{\theta^{(\ell)}}(\cdot) : \mathbb{R}^{d \times s} \to \mathbb{R}^{d \times s}$ describe the sequence-to-sequence function mapping with parameters $\theta^{(\ell)}$ that mainly comprises two complementary stages of data transformation: the Multi-head Self-Attention across tokens and the MultiLayer Perceptron layer across features. Predictions are typically based on a specific token of last layer representation $\boldsymbol{H}^{(L)}$. For instance, ViT uses the representation of a class token [CLS] to classify the image, while auto-regressive based language model (such as GPT) uses the representation of the previous tokens to predict the next word. Consequently, we will focus on the feature (or representation) of this particular token in $\boldsymbol{H}^{(\ell)}$, denoted by $\boldsymbol{h}^{(\ell)} \in \mathbb{R}^d$ at the $\ell$-th layer. A linear classifier $g(\cdot)$ is applied to the last layer feature

$h^{(L)}$ to make predications as[1] $g(h^{(L)}) = \arg\max_j[\text{SoftMax}(Wh^{(L)})]_j$, where $[\cdot]_j$ denotes the $j$-th entry and $W \in \mathbb{R}^{K \times d}$ maps the $d$ dimensional features to $K$ dimensional logits. We may also directly apply the last layer classifier $g(\cdot)$ to the hidden layer features $h^{(\ell)}$ to make predictions via $g(h^{(\ell)}) = \arg\max_j[\text{SoftMax}(Wh^{(\ell)})]_j$. Given data samples $\mathcal{S} := \{x_{k,i}\}$, where $x_{k,i}$ represents the $i$-th sample of class $k$ with $i \in [n] := \{1, \ldots, n\}$ and $k \in [K]$, we define layer-wise accuracy as

$$\text{Acc}_{\mathcal{S}}^{(\ell)} := \frac{1}{Kn} \sum_{k=1}^{K} \sum_{i=1}^{n} \mathbb{1}[g(h_{k,i}^{(\ell)}) = k]. \tag{2}$$

**Existing work on measuring representational similarity**  Similarity analysis is widely applied in the literature, including research on learning dynamics (Morcos et al., 2018; Mehrer et al., 2018), effects of width and depth (Nguyen et al., 2020), differences between supervised and unsupervised models (Gwilliam & Shrivastava, 2022), robustness (Jones et al., 2022; Nanda et al., 2022), evaluating knowledge distillation (Stanton et al., 2021), language representation (Kudugunta et al., 2019; Shi et al., 2022), and generalizability (McCoy et al., 2019; Lee et al., 2022; Pagliardini et al., 2022). To enable measuring the similarity of features from from different architectures or layers that have different dimension, most existing methods for analyzing the similarity between representations of high dimensions, such as those based on Canonical Correlation Analysis (CCA) and widely used Centered Kernel Alignment (CKA) (Kornblith et al., 2019) , rely on statistical properties of the representations for a set of data points. For instance, given input feature sequence $Z^{\ell} = \begin{bmatrix} h_{1,1}^{(\ell)} & \cdots & h_{K,n}^{(\ell)} \end{bmatrix} \in \mathbb{R}^{d \times N}$ denoting the features of $N = Kn$ training samples, the widely-used CKA with a linear kernel quantifies similarities between features $Z^{\ell}$ and $Z^{\ell'}$ as

$$\text{CKA} = \text{Tr}((Z^{\ell'})^{\top} Z^{\ell'} \cdot (Z^{\ell})^{\top} Z^{\ell}) / (\|Z^{\ell}(Z^{\ell})^{\top}\|_F \|Z^{\ell'}(Z^{\ell'})^{\top}\|_F). \tag{3}$$

CKA relies on the similarity of inter-example structures since the gram matrix $(Z^{\ell})^{\top} Z^{\ell} \in \mathbb{R}^{N \times N}$ captures the pair-wise similarity of different samples, focusing on the consistency of relative positions among features. Consequently, the CKA is invariant to orthogonal transformations and isotropic scaling, and can be applied for the case where $h^{(\ell)}$ and $h^{(\ell')}$ have different dimensions.

## 2.1 Sample-wise Layer-wise Representational Similarity in Transformers

In this work, we specifically focus on transformer architectures that obey the following particular properties of features across layers: $(i)$ the features have the same dimension across layers since transformers are generally constructed by stacking identical blocks; $(ii)$ the features may have no or less rotation ambiguity due to the residual connection (1). Based on these observations, for each sample $x_{k,i}$ we propose to simply measure the cosine similarity of the corresponding feature vectors $h^{(\ell)}$ and $h^{(\ell')}$ at layers $\ell$ and $\ell'$ as[2]

$$\text{COS}(h_{k,i}^{(\ell)}, h_{k,i}^{(\ell')}) = \langle h_{k,i}^{(\ell)}, h_{k,i}^{(\ell')} \rangle / \|h_{k,i}^{(\ell)}\|_2 \|h_{k,i}^{(\ell')}\|_2.$$

The above COsine Similarity (COS) measures the angle between feature vectors, providing a clear geometric interpretation of feature alignment and similarity at the layer level. Unlike CKA (3), COS is not invariant to all transformations except for isotropic scaling. Furthermore, COS is computed for each individual sample and does not rely on inter-example structures. In the experiments, we compute the average COS over all the training samples.

To verify whether the proposed sample-wise COS is a good indicator of similarity structure within transformers, we train the DeiT-S model (Touvron et al., 2021) (a data-efficient vision transform) from scratch on both the CIFAR-10 and ImageNet-1K datasets. The feature dimension is set to 384 for both tasks across all layers. In Figure 2(a, b), we compute CKA and average cosine similarity between the features in each layer and the last layer. Additionally, we plot average COS between all pairs of layers and display the results as a heatmap in Figure 2(c, d). Based on these results, we make several observations.

---

[1] There is also a bias term $b$ in classification layer, but we omit it for simplicity of presentation.
[2] The features in each layer are centered by reducing the global mean of all the samples.

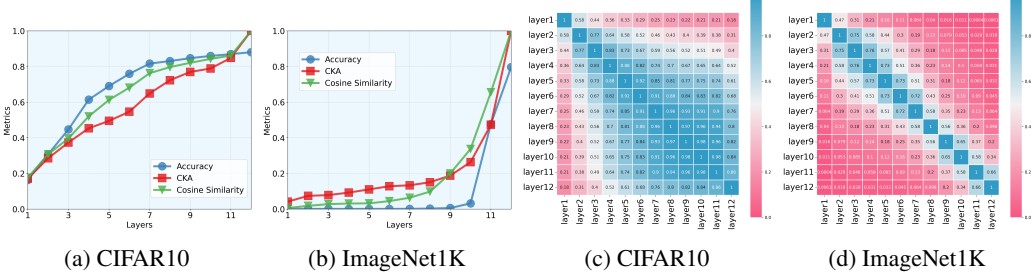

| (a) CIFAR10 | (b) ImageNet1K | (c) CIFAR10 | (d) ImageNet1K |

Figure 2: Illustration of a DeiT-S model trained with standard training on CIFAR-10 and ImageNet in terms of (a-b) similarity of features from shallow layers and the last-hidden layer measured by CKA and COS, as well as layerwise validation accuracy, and (c-d) cosine similarities between all pairs of layers. For both datasets, cosine similarity can reflect the trend of layerwise accuracy.

**Observation 1: Simple-wise COS is sufficient and reflects CKA to measure layer-wise representation similarity** We observe from Figure 2(a, b) that COS aligns with CKA and is sufficient to reflect the layer-wise representation similarity in transformer models. In other words, since CKA is invariant to orthogonal transformations while COS is not, there is little or no rotation difference among the features from different layers. This is attributed to the residual connections in (1), which create smoother transitions between layers and potentially lead to more stable feature representations throughout the network. In the Appendix B.2, we design additional experiments on multi-layer perceptrons (MLPs) with and without skip connections to verify the effect of skip connections in eliminating rotation ambiguity.

**Observation 2: Progressively increasing layer-wise representation similarity.** For models trained on small datasets, we observe a ridge-to-plateau pattern in the heatmap plot in Figure 2(c): the initial layers undergo significant transformations, suggesting that lower layers rapidly refine the features to extract the most relevant information for classification tasks; in contrast, the higher layers exhibit a plateau in similarity scores, indicating that feature transformations stabilize and converge toward an optimal representation. This plateau also suggests redundancy in the higher layers, implying that removing these redundant layers could improve efficiency without significantly sacrificing performance.[3] On the other hand, for models trained on large datasets, Figure 2(d) shows a consistent ridge pattern, characterized by a rapid decay in similarity between adjacent layers. This indicates a dynamic and continuous refinement process throughout the network. Nevertheless, for both Figure 2(c, d), we observe almost all nonnegative average cosine similarity between different layers, albeit the features are almost orthogonal when the layers are far apart. Moreover, in both Figure 2(c, d), we observe a progressive increase in representation similarity as the two layers get closer; across each row or column, the cosine similarity increases as it approaches the diagonals. In appendix D, we observe similar phenomena on multi-modality models (CLIP).

To understand this phenomenon, we utilize the connection between residual network (ResNet) and dynamic system, viewing the residual update (1) as a discretization of a dynamic system (Haber & Ruthotto, 2017; Gai & Zhang, 2021). Specifically, Gai & Zhang (2021) proved that ResNet trained with weight decay attempts to learn the geodesic curve in the Wasserstein space. Since transformer is a ResNet and weight decay is commonly applied in real-world training—for example, DeiT (Touvron et al., 2021) models are trained using the AdamW optimizer with a weight decay of 0.05—we use the following geodesic curve assumption (Wang et al., 2024).

**Assumption 1.** *(Geodesic curve assumption) At the terminal phase of training, the transformer with weight decay has learned the geodesic curve in Wasserstein space $\mathcal{P}(\mathbb{R}^d)$, which is induced by the optimal transport map.*

Based on this assumption, the following result shows a monotonic increase in representation similarity as the layers get closer. Proofs are given in the Appendix A.

**Theorem 1.** *(Representation similarity increases monotonically across layers) Under Assumption 1, for any layers $\ell_1 < \ell_2 < \ell_3$, we have $\mathrm{COS}(\boldsymbol{h}_{k,i}^{(\ell_1)}, \boldsymbol{h}_{k,i}^{(\ell_3)}) < \mathrm{COS}(\boldsymbol{h}_{k,i}^{(\ell_2)}, \boldsymbol{h}_{k,i}^{(\ell_3)})$.*

---

[3]We notice concurrent work (Men et al., 2024; Gromov et al., 2024; Jaiswal et al., 2024) that exploits representation similarity across layers for pruning redundant layers. For instance, Figure 2(c, d) show hidden layers obey large similarity, indicating that some of the layers can be skipped (Jaiswal et al., 2024).

Theorem 1 provides a theoretical justification for the phenomenon observed in Figure 2—for instance, the similarity to the last layer features progressively increases from shallow to deep layers. However, we note that in practice, the geodesic curve assumption may not hold precisely by a practical network, so some samples may not exhibit a strictly monotonic increase.

## 2.2 ANALYSIS FOR SATURATION EVENTS

We now use the layer-wise representation similarity to analyze the phenomenon of *saturation events* (Geva et al., 2022). To that goal, we first briefly introduce the neural collapse ($\mathcal{NC}$) phenomenon (Papyan et al., 2020) and its connection to layer-wise representation similarity.

**Neural Collapse ($\mathcal{NC}$)** Roughly speaking, $\mathcal{NC}$ concerns the terminal phase of training deep networks and states that $(i)$ within-class variable collapse ($\mathcal{NC}_1$): the last-layer features from the same class become nearly identical, i.e., $\boldsymbol{h}_{k,i}^{(L)} \to \overline{\boldsymbol{h}}_k^{(L)} = \frac{1}{n}\sum_i \boldsymbol{h}_{k,i}^{(L)}$, $(ii)$ maximal distance ($\mathcal{NC}_2$): those from different classes become maximally linearly separable, and $(iii)$ self-duality ($\mathcal{NC}_3$): the last-layer linear classifiers $\boldsymbol{w}_k$ align with the class-mean features $\overline{\boldsymbol{h}}_k^{(L)}$. To achieve this, deep classifiers progressively compress within-class features while enhancing the discrimination of between-class features from shallow to deep layers (He & Su, 2023; Rangamani et al., 2023; Wang et al., 2023). Our results show that transformer models achieve progressive compression and separation by progressively aligning the features with the last-layer classifier from shallow to deep layers.

**Alignment between layer-wise cosine similarity and accuracy.** Motivated by the similarity between features in shallow and deep layers, we apply the classifier to each hidden layer to obtain the layer-wise validation accuracy, which is plotted in Figure 2(a, b). We observe a high correlation between the layer-wise accuracy and the cosine similarity, with layer-wise accuracy also exhibiting a monotonic increase across layers. Our following result provides a justification for this phenomenon, demonstrating that an increase in representation similarity implies an increase in predicted probability across layers.

**Theorem 2.** *(Predicted probability increases monotonically across layers) Assume that Theorem 1 holds at $\boldsymbol{h}_{k,i}$, i.e., $\mathrm{COS}(\boldsymbol{h}_{k,i}^{(\ell+1)}, \boldsymbol{h}_{k,i}^{(L)}) > \mathrm{COS}(\boldsymbol{h}_{k,i}^{(\ell)}, \boldsymbol{h}_{k,i}^{(L)})$, and that the last-layer features and classifers satisfy $\mathcal{NC}$. Then, the predicted probability $[SoftMax(\boldsymbol{W}\boldsymbol{h}_{k,i}^{(\ell)})]_k$ increases across layers:*

$$[SoftMax(\boldsymbol{W}\boldsymbol{h}_{k,i}^{(\ell+1)})]_k > [SoftMax(\boldsymbol{W}\boldsymbol{h}_{k,i}^{(\ell)})]_k. \tag{4}$$

**Saturation events** The approach of applying last-layer classification to intermediate representation is also called logit lens (nostalgebraist). Recent studies (Belrose et al., 2023; Pal et al., 2023) use this method to decode hidden states into probability distributions over the vocabulary, offering mechanistic interpretability of transformers. (Geva et al., 2022) discover a phenomenon called saturation events, where the model's final predicted token becomes the top candidate in a certain shallow layer and remains unchanged across all subsequent layers. Specifically, given an input sample $\boldsymbol{x}_{k,i}$, the saturation layer $\ell_{k,i}$ for $\boldsymbol{x}_{k,i}$ is defined as the smallest layer $\ell$ such that

$$g(\boldsymbol{h}_{k,i}^{(1)}) \neq \ldots \neq g(\boldsymbol{h}_{k,i}^{(\ell)}) = \cdots = g(\boldsymbol{h}_{k,i}^{(L)}).$$

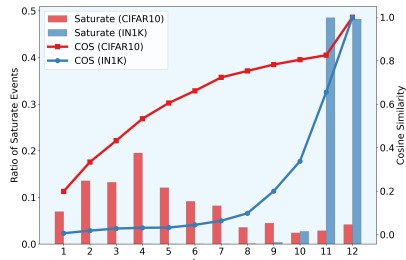

Figure 3: The DeiT-S models are trained on CIFAR10 and Imagenet-1K dataset from scratch. We measure the number of saturate event at each layer and their average cosine similarity with last hidden states. More saturate events at early layer indicates higher cosine similarity.

Results in Appendix C.1 show saturate events also happen on recently developed LLMs such as LLaMA3 (Dubey et al., 2024). Our Theorem 2 offers a justification for saturation events, stating that if a sample is correctly predicted at the $\ell$-th layer, it will continue to be correctly predicted in subsequent layers, as the predicted probability increases progressively across layers. To further illustrate the relation between representation similarity and saturate events, we train a DeiT-S model on both CIFAR-10 and ImageNet-1K and plot the results in Figure 3. For the same model across different datasets, we observed that higher layer-wise representation similarity COS correlates with more early saturation events, suggesting COS is a valuable metric for reflecting saturate events.

Table 1: Comparison of the number of parameters across different architectures between multi-exit models with multiple classifiers (different classifiers at each layer) and a single classifier (ours). The last column shows that the percentage of saved parameters using single classifier.

| Models | Hidden Dim | # of Classes | # of Layers | Multiple Classifiers (#Params) | Single Classifier (#Params) | #Param Saving |
|---|---|---|---|---|---|---|
| DeiT-S(Touvron et al., 2021) | 384 | 1,000 | 12 | 26.27M | 22.05M | 16.07% |
| DeiT-B(Touvron et al., 2021) | 768 | 1,000 | 12 | 95.02M | 86.57M | 8.89% |
| GPT-2(Radford et al., 2019) | 768 | 50,257 | 12 | 541.57M | 117.35M | 78.39% |
| GPT-3(Brown et al., 2020) | 12,288 | 50,257 | 96 | 233.67B | 175.63B | 25.10% |
| LLAMA-2 (Touvron et al., 2023) | 4,096 | 32,000 | 40 | 75.11B | 70.35B | 6.81% |

## 3  ALIGNED TRAINING FOR ENHANCING REPRESENTATIONAL SIMILARITY

In the previous section, we observed that representations across layers within transformer models are positively correlated, resulting in saturation events when the last-layer classifier is directly applied after any hidden layer for early prediction and enabling a multi-exit model that shares a single classifier. In this section, we propose an aligned training method to enhance the effectiveness of shallow layers by improving layer-wise feature similarity. This, in turn, promotes more early saturation events, determines the minimal effective depth, and enhances performance when used as a multi-exit model. To the best of our knowledge, our work is the first to show that one common classifier is sufficient for multi-exit models. Table 1 shows a single classifier can significantly reduce the number of parameters and the computational complexity for multi-exit models, particularly for tasks with a large number of classes and large feature dimensions. Examples include ImageNet, with 1000 classes, and LLMs, where the number of classes equals the vocabulary size, i.e., the number of all possible tokens—for instance, the Llama-2 (Touvron et al., 2023) model has a vocabulary of $32,000$ tokens while the GPT3 (Brown et al., 2020) has $50,257$ tokens.

### 3.1  ALIGNED TRAINING FOR ENHANCING SHALLOW LAYER PERFORMANCE

The ability to capture the layer-wise similarity of representations for each sample enables us to develop efficient methods for enhancing this similarity during training. A first approach is to directly add the cosine similarity between $\boldsymbol{h}^{(\ell)}$ and $\boldsymbol{h}^{(L)}$ for all $\ell < L$ as a regularization term during the training process. However, as shown in the appendix (see Figure 14), this approach can only slightly improve layer-wise similarity and accuracy. We conjecture this is due to the imbalance between the cross-entropy loss and the cosine similarity. Instead, motivated by the self-duality between the class-mean features and the linear classifiers, as observed in the $\mathcal{NC}$ phenomenon, we propose a simple yet efficient method, named aligned training, to enhance the layer-wise similarity by jointly optimizing the following aligned loss that is the weighted average of the CE loss from all the layers

$$\mathcal{L}_{\text{aligned}}(\boldsymbol{x}, \boldsymbol{y}) = \sum_{\ell=1}^{L} \lambda_\ell \mathcal{L}_{\text{CE}}(\boldsymbol{W}\boldsymbol{h}^{(\ell)}, \boldsymbol{y}), \tag{5}$$

where $\boldsymbol{y}$ denotes the corresponding label for $\boldsymbol{x}$, $\lambda_\ell > 0$ is the weight for the $\ell$-th layer. During the experiments, considering that shallow layers tend to have larger losses compared to deep layers, we set the weight to linearly increase with layers to put more emphasis on the deeper layers[4], i.e., $\lambda_\ell = 2\ell/(L(L+1))$. Roughly speaking, the aligned loss (5) introduces CE loss for intermediate layers and would encourage each layer features $\boldsymbol{h}^{(\ell)}$ to align with the common classifier $\boldsymbol{W}$—as implied by the $\mathcal{NC}$ phenomenon—hence improving the representation similarity across layers.

Figure 4 displays the layer-wise representation similarity and accuracy by the proposed aligned training. We can observe that **aligned training can significantly increase the layer-wise representation similarity and accuracy** by aligning all the features to the common classifier. Results in Appendix B.3 also show that aligned training can enhance progressive separation and compression from shallow to deep layers. To further illustrate the benefit of aligned training, we define the notion of $\epsilon$-effective depth that modifies the notion exploited in Galanti et al. (2022) by replacing nearest neighbor classifier accuracy with our layer-wise accuracy in (2).

---

[4]Such a strategy of increasing weights is also employed in (Schuster et al., 2022). Uniformly weighting all layers (i.e., $\lambda_\ell = 1/L$) may diminish the importance of the deeper layers. We provide an ablation study for the comparison of linear increasing weights and uniform weights in Appendix B.3.

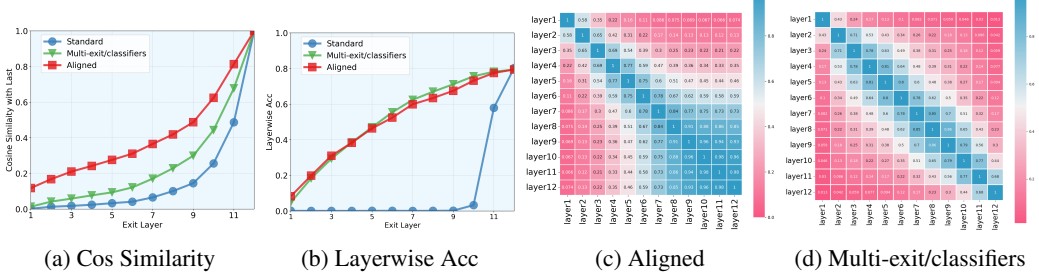

| (a) Cos Similarity | (b) Layerwise Acc | (c) Aligned | (d) Multi-exit/classifiers |

Figure 4: Comparison of ViT for ImageNet by standard training, proposed aligned training, and the multi-exit/classifiers, in terms of (a) cosine similarity, (b) layer-wise testing accuracy and (c-d) cosine similarities between all pairs of layers.

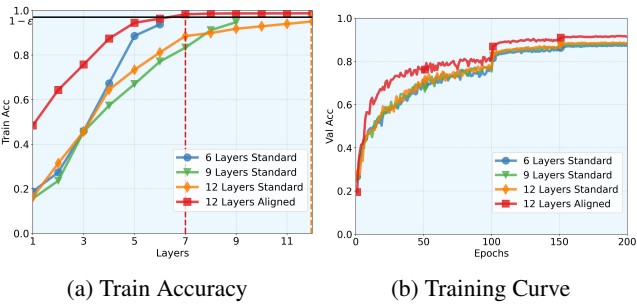

| (a) Train Accuracy | (b) Training Curve |

Figure 5: Comparison of standard training of 6, 9, 12-layer DeiT-small model with aligned training of 12-layer model on CIFAR-10 in terms of (a) layer-wise train accuracy, and (b) convergence.

**Definition 1.** *($\epsilon$-effective depth). We define the $\epsilon$-effective depth $d^\epsilon(\mathcal{S}, f)$ of an L-layer transformer $f$ over dataset $\mathcal{S}$ as the minimal layer $\ell$ such that $Acc_{\mathcal{S}}^{(\ell)} \geq 1 - \epsilon$. Set $d^\epsilon(\mathcal{S}, f) = L$ if such $\ell$ is non-existent.*

**Minimal $\epsilon$-effective depth**    Denote the transformers learned by standard training and aligned training as $f_{\text{standard}}$ and $f_{\text{aligned}}$, respectively. We observe that the aligned training yields a model with a much smaller $\epsilon$-effective depth compared to standard training, i.e., $d^\epsilon(\mathcal{S}, f_{\text{aligned}}) < d^\epsilon(\mathcal{S}, f_{\text{standard}})$. Specifically, Figure 5(a) displays the layer-wise training accuracy of three transformers with different number of layers $\ell \in [6, 9, 12]$ with standard training and a 12-layer transformer with aligned training. For the standard training models, the train accuracy curves increase until the last layer without plateauing, even for the model with 12 layers, giving $d^\epsilon(\mathcal{S}_{\text{CIFAR10}}, f_{\text{standard}}) = 12$. While aligned training models unleash the power of shallow layers to transform features faster towards classifier across layers and push the redundancy behind, giving $d^\epsilon(\mathcal{S}_{\text{CIFAR10}}, f_{\text{aligned}}) \approx 7$, which is smaller than $\epsilon$-effective depth of $f_{\text{standard}}$. On the other hand, the effective depth $d^\epsilon(\mathcal{S}, f_{\text{aligned}})$ increases with the complexity of the task, as demonstrated by comparing the results from CIFAR10 (Figure 5(a)) and ImageNet (Figure 4(b)). Thus, the effective depth, independent of the network's depth, can be leveraged to derive generalization bounds. This can be achieved by applying the approach from Galanti et al. (2022), which offers non-trivial estimates of generalization based on effective depth.

Models $f_{\text{aligned}}$ with small effective depth also offer several advantages in practical deployment. First, the layers beyond effective depth $d^\epsilon(\mathcal{S}_{\text{CIFAR10}}, f_{\text{aligned}})$ are redundant, as they do not contribute to accuracy improvements and can be pruned, leading to more efficient inference. Second, aligned training helps determine the minimal number of layers required for a task. While more complex tasks typically demand more layers, identifying the exact number can be challenging. In standard training, multiple models of different sizes must be trained to determine the minimal layer count. In contrast, with aligned models, retraining is unnecessary—one can simply apply the last-layer linear classifier to intermediate layers. Third, models trained using aligned method not only achieve slightly higher accuracy when truncated to 6 or 9 layers compared to models of the same size trained with standard methods (Figure 5(a)), but they also accelerate model convergence (Figure 5(b)) by providing immediate feedback to each layer, resulting in more effective parameter adjustments.

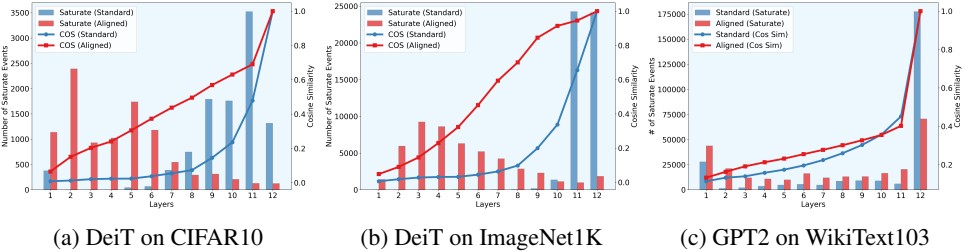

|  (a) DeiT on CIFAR10 | (b) DeiT on ImageNet1K | (c) GPT2 on WikiText103 |

Figure 6: Illustration of the effect of aligned training versus standard training on cosine similarity and the number of saturation events across vision (a,b) and NLP (c) models. Aligned training leads to more early saturation events by increasing the cosine similarity with the last hidden states.

**Saturation events and multi-exit model with a single classifier** Another important application of the proposed aligned training is to improve the inference efficiency of large models. Figure 6 illustrates saturation events in both vision and NLP models (see the AlignedGPT setup in the appendix). We observe that aligned training **encourages earlier saturation events** by increasing the cosine similarity with the final hidden states. This demonstrates that $f_{\text{aligned}}$ has greater potential for supporting early exits, commonly referred to as multi-exits in the literature (Xin et al., 2020; Geng et al., 2021; Xin et al., 2021). In previous work, multi-exit models typically use separate classifiers for each layer, which can significantly increase the overall model size. As shown in Table 1, this issue becomes more pronounced when dealing with a large number of classes, as dense linear classifiers require a substantial number of parameters. To our knowledge, by exploiting the representation similarity within the transformer, *our proposed multi-exit model is the first to use a single classifier for all the layers*. In addition, when training multi-exit models, previous work (Geng et al., 2021) often uses additional KL-divergence terms to guide the logits of shallow layers by those of deep layers. In contrast, by using a common classifier to align shallow representations with deep ones, our aligned training does not require KL-divergence or other such terms. For comparison, we implement the multi-exit training with multiple classifiers (Xin et al., 2021), denoted by "multi-exit/classifiers", and display the results in Figure 4. On one hand, we observe that the proposed aligned training with a shared classifeir achieves higher layer-wise representation similarity than multi-exit/classifiers. On the other hand, the aligned training exhibits on-par performance as multi-exit/classifiers in terms of layer-wise accuracy, which is remarkable as the former only uses a single classifier.

To further show the performance of the proposed multi-exit models with a single classifier, we allow exit at shallow layers if the confidence level (max of softmax logits) exceeds a set threshold for each sample. On ImageNet dataset, Figure 7 displays the number of samples that exit at each layer. We observe that most samples exit at the last layer for standard training, while most exit at early layers for aligned training. We then calculate the classification accuracy along with the ratio of speed improvement measured by $\frac{\sum_{i=1}^{L} L \times m^i}{\sum_{i=1}^{L} i \times m^i}$ where $m^i$ is the number of samples that exit at the $i$-th layer of DeiT. The model trained with standard training achieves 80.28% accuracy, while the model

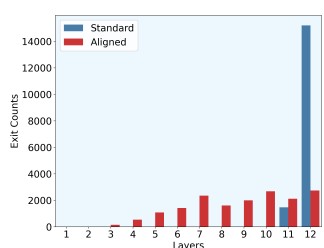

Figure 7: Number of samples exit at each layer.

trained with aligned training achieves 77.96% accuracy with a $1.36\times$ speedup, which is comparable to those trained by multi-exit/classifiers with 78.32 % accuracy and $1.42\times$ speedup.

**Effects on Transferability** Concerns arise about whether aligning shallow layer features with deep layer features could diminish the transferability of shallow layers. To answer this question, we conduct experiments on (1) distribution shift and (2) task transfer. The results show that aligned training improves layer-wise accuracy for both pre-trained and downstream datasets while maintaining transferability. This indicates that the trained model can be effectively transferred without losing the universal patterns learned in shallow layers. Further details are provided in Appendix B.4.

### 3.2 APPLICATIONS ON LANGUAGE MODELS

We extend our aligned training approach to NLP tasks, demonstrating its effectiveness in fine-tuning Language Models. For text classification tasks, we get **AlignedBERT** by finetuning a pretrained 12-

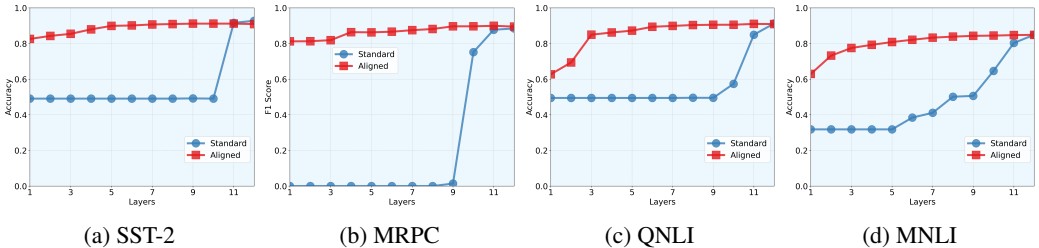

Figure 8: Layerwise accuracy for **AlignedBERT** and BERT and on SST-2, MRPC, QNLI and MNLI datasets of the GLUE benchmark. (See more results on RTE and QQP in Figure 21).

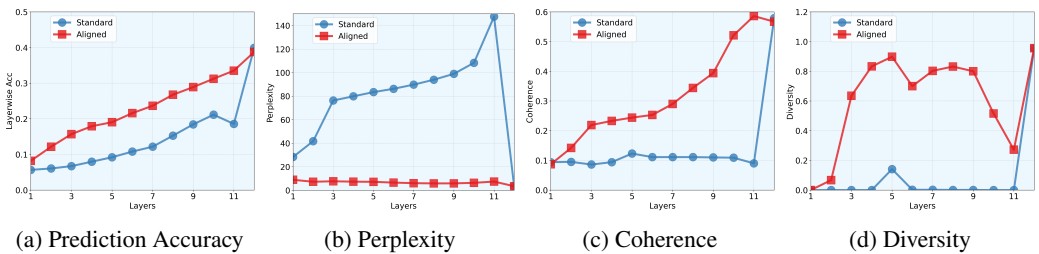

Figure 9: Evaluation of **AlignedGPT** and GPT2 on Wikitext-103 dataset in terms of (a) prediction accuracy, (b) perplexity, (c) coherence, and (d) diversity. See Appendix C.2 for detailed definitions.

layer BERT$_{Base}$ model (Devlin, 2018) using aligned training method on GLUE benchmark (Wang et al., 2018) tasks. This includes single-sentence tasks-SST-2, similarity and paraphrasing tasks-MRPC and QQP, as well as natural language inference tasks-MNLI, QNLI and RTE. For comparison, we also finetune a BERT$_{Base}$ model using standard training. Then we evaluate the layerwise accuracy(or F-1 score) of both finetuned models using their last layer classifier. Figure 8 shows that AlignedBERT achieves better performance than the standard BERT across all layers.

For text generation task, we get **AlignedGPT** by finetuning a pretrained 12-layer GPT2 model (Radford et al., 2019) using aligned training method on Wikitext-103 dataset (Merity et al., 2016) . For comparison, we also finetune a GPT2 model using standard training. Then we evaluate the two finetuned models from two perspectives(following (Su et al., 2022; Su & Collier, 2023)), (1) language modeling quality, which assesses the intrinsic quality of the model and is measured by prediction accuracy and perplexity, and (2) generation quality, which measures the quality of the text produced by the model using coherence and diversity. Coherence is a measurement of relevance between prefix text and generated text, while diversity considers the recurrence of generation at varying n-gram levels. All evaluation metrics mentioned above can be found in Appendix C.2. Results show that AlignedGPT outperforms the standard one in prediction accuracy and exhibits lower perplexity across intermediate layers(Figure 9(a,b)). Moreover, AlignedGPT can also generate text with higher coherence and diversity using shallower layers(Figure 9(c,d)), which improves the the inference efficiency. More experimental setup and results can be found in Appendix C.2.

## 4 CONCLUSION

In this paper, we demonstrate that a simple sample-wise cosine similarity metric effectively captures layer-wise representation similarity in transformer models, aligning with the more complex CKA metric. Our findings reveal that representations become more similar as layers get closer and show that increased representation similarity correlates with higher prediction accuracy, leading to saturation events where shallow layers can already make correct predictions. To enhance this, we proposed an aligned training method that improves shallow layer effectiveness, resulting in more early saturation events and much higher layer-wise accuracies. Remarkably, when served as multi-exit models with a common classifier, which to our knowledge is the first of its kind, they maintain the early exit capability and achieve performance on par with models that use multiple classifiers. Experiments on both vision and NLP tasks demonstrate the performance of the proposed aligned training.

ACKNOWLEDGEMENT

We acknowledge support from NSF grants CCF-2240708, IIS-2312840, and IIS-2402952, as well as the ORAU Ralph E. Powe Junior Faculty Enhancement Award. We are grateful to Beidi Chen (Carnegie Mellon University) and Chong You (Google Research) for many valuable discussions and for helpful comments on the manuscript.

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

# Appendix

**Notations and Organizations.** The appendix provides theorem proofs as well as additional experimental results on both vision and language models. It is organized as follows: First, we present the proofs for the theorems in Appendix A. Next, we provide additional experiment results on vision models(Appendix B), language models(Appendix C) and multi-modality models(Appendix D).

## A  PROOF FOR THEOREMS

In this section, we provide proof for the theorem 1 and 2.

### A.1  PROOF FOR THEOREMS 1

As a result of the geodesic curve assumption, the forward propagation of transformer is along a straight line in the feature space $\mathbb{R}^d$. Formally, the feature $\boldsymbol{h}_{k,i}^{(\ell)}, \ell \in [0, L]$ is along the line $\boldsymbol{h}_{k,i}^{(\ell)} = (1 - \frac{\ell}{L})\boldsymbol{h}_{k,i}^{(0)} + \frac{\ell}{L}\boldsymbol{h}_{k,i}^{(L)}$. In the following proof, we drop the subscript $k, i$ in $\boldsymbol{h}$.

Let $x = \frac{\ell}{L}$, so $x \in [0, 1]$. Suppose the feature of the first layer is $\boldsymbol{h}^{(0)}$ and feature of the last layer is $\boldsymbol{h}^{(1)}$. Then:

$$\boldsymbol{h}^{(x)} = (1 - x)\boldsymbol{h}^{(0)} + x\boldsymbol{h}^{(1)} \tag{6}$$

The cosine of the angle between intermediate layer feature $\boldsymbol{h}^{(x)}$ and last layer feature $\boldsymbol{h}^{(1)}$ is,

$$C(x) = \cos(\boldsymbol{h}^{(x)}, \boldsymbol{h}^{(1)}) = \frac{\langle \boldsymbol{h}^{(x)}, \boldsymbol{h}^{(1)} \rangle}{\|\boldsymbol{h}^{(x)}\|\|\boldsymbol{h}^{(1)}\|} \tag{7}$$

Since $\|\boldsymbol{h}^{(1)}\|$ is fixed, we can treat $\|\boldsymbol{h}^{(1)}\|$ as constant. For simplicity, define $N(x) = \langle \boldsymbol{h}^{(x)}, \boldsymbol{h}^{(1)} \rangle$ and $D(x) = \|\boldsymbol{h}^{(x)}\|$. Thus, we can get,

$$C(x) = \frac{\langle \boldsymbol{h}^{(x)}, \boldsymbol{h}^{(1)} \rangle}{\|\boldsymbol{h}^{(x)}\|\|\boldsymbol{h}^{(1)}\|} = K\frac{N(x)}{D(x)} \tag{8}$$

where $K = \frac{1}{\|\boldsymbol{h}^{(1)}\|}$ is a positive constant. Note that

$$N(x) = \langle \boldsymbol{h}^{(x)}, \boldsymbol{h}^{(1)} \rangle = (1 - x)\langle \boldsymbol{h}^{(0)}, \boldsymbol{h}^{(1)} \rangle + x\|\boldsymbol{h}^1\|^2$$

$$D(x) = \|\boldsymbol{h}^{(x)}\| = \sqrt{(1 - x)^2\|\boldsymbol{h}^0\|^2 + 2x(1 - x)\langle \boldsymbol{h}^{(0)}, \boldsymbol{h}^{(1)} \rangle + x^2\|\boldsymbol{h}^1\|^2}$$

To prove $C(x)$ monotonically increase within $x \in [0, 1]$, we can take derivative of $C(x)$ with respect to $x$ and show $\frac{dC(x)}{dx} > 0$ for $x \in [0, 1]$. The derivative is given by

$$\frac{dC(x)}{dx} = K\frac{\frac{dN(x)}{dx}D(x) - \frac{dD(x)}{dx}N(x)}{D^2(x)} \tag{9}$$

Assuming $\boldsymbol{h}^{(0)}$ and $\boldsymbol{h}^L$ are unit vectors( $\|\boldsymbol{h}^{(0)}\| = \|\boldsymbol{h}^{(1)}\| = 1$) and defining $c = \langle \boldsymbol{h}^{(0)}, \boldsymbol{h}^{(1)} \rangle$ (which satisfies $-1 \leq c \leq 1$), we have:

$$N(x) = (1 - x)c + x = (1 - c)x + c$$

$$D(x) = \sqrt{(1 - x)^2 + 2x(1 - x)c + x^2}$$

Taking derivative of both $N(x)$ and $D(x)$ with respect to $x$ gives

$$\frac{dN(x)}{dx} = (1 - c)$$

$$\frac{dD(x)}{dx} = \frac{(2x - 1)(1 - x)}{D(x)}$$

Then we only need to check the sign of $\frac{dN(x)}{dx}D(x) - \frac{dD(x)}{dx}N(x)$ term,

$$\frac{dN(x)}{dx}D(x) - \frac{dD(x)}{dx}N(x) = (1-c)D(x) - \frac{(2x-1)(1-x)}{D(x)}((1-c)x+c)$$

$$= \frac{1}{D(x)}((1-c)D^2(x) - (2x-1)(1-x)((1-c)x+c))$$

$$= \frac{1}{D(x)}(2cx^2 - (1+3c)x + (1+c))$$

where $x \in [0,1]$ and $c \in [-1,1]$. Noticing that $\frac{1}{D(x)} \geq 0$, we define

$$P(x) = 2cx^2 - (1+3c)x + (1+c) \tag{10}$$

We have $P(0) = 1 + c \in [0,2]$ and $P(1) = 0$. If $P(x)$ is negative between $[0,1]$, for a quadratic function, there is only one possibility: the axis of symmetry $x = \frac{1+3c}{4c}$ is between 0 and 1, and the parabola opens upward:

$$\begin{cases} 0 < \frac{1+3c}{4c} < 1 \\ c > 0 \\ -1 < c < 1 \end{cases} \tag{11}$$

These conditions are contradictory, and no value of $c$ satisfies all of them simultaneously. Thus, $P(x) \geq 0$ always holds for $x \in [0,1]$. Consequently, $C(x) = \cos(\boldsymbol{h}^{(x)}, \boldsymbol{h}^{(1)})$ increases monotonically for $x \in [0,1]$. Thus, for any layers $\ell_1 < \ell_2 < \ell_3$, the relationship $\cos(\boldsymbol{h}^{(\ell_1)}, \boldsymbol{h}^{(\ell_3)}) < \cos(\boldsymbol{h}^{(\ell_2)}, \boldsymbol{h}^{(\ell_3)})$ holds true.

### A.2 PROOF FOR THEOREM 2

According to the COsine Similarity (COS) improvement, $\text{COS}(\boldsymbol{h}_{k,i}^{(\ell+1)}, \boldsymbol{h}_{k,i}^{(L)}) > \text{COS}(\boldsymbol{h}_{k,i}^{(\ell)}, \boldsymbol{h}_{k,i}^{(L)})$. By $\mathcal{NC}_1$, we have $\text{COS}(\overline{\boldsymbol{h}}_k^{(\ell+1)}, \overline{\boldsymbol{h}}_k^{(L)}) > \text{COS}(\overline{\boldsymbol{h}}_k^{(\ell)}, \overline{\boldsymbol{h}}_k^{(L)})$; by $\mathcal{NC}_3$, we have $\text{COS}(\overline{\boldsymbol{h}}_k^{(\ell+1)}, \boldsymbol{w}_k^{(L)}) > \text{COS}(\overline{\boldsymbol{h}}_k^{(\ell)}, \boldsymbol{w}_k^{(L)})$. Assume the norm of $\boldsymbol{h}_k^{(\ell)}$ and $\boldsymbol{h}_k^{(\ell+1)}$ are equal, which is $\|\boldsymbol{h}_k^{(\ell)}\| = \|\boldsymbol{h}_k^{(\ell+1)}\| = h$, so we have,

$$\boldsymbol{w}_k^T \boldsymbol{h}_k^{(\ell+1)} > \boldsymbol{w}_k^T \boldsymbol{h}_k^{(\ell)} \tag{12}$$

Suppose that $\boldsymbol{h}_k^{(\ell+1)} = \boldsymbol{h}_k^{(\ell)} + \Delta \boldsymbol{h}$, so we have,

$$\boldsymbol{w}_k^T \Delta \boldsymbol{h} > 0 \tag{13}$$

The logits for class $k$ at layer $\ell$ and layer $\ell+1$ are denoted by $z_k^{(\ell)} = \boldsymbol{w}_k^T \boldsymbol{h}_k^{(\ell)}$ and $z_k^{(\ell+1)} = \boldsymbol{w}_k^T \boldsymbol{h}_k^{(\ell+1)}$, respectively. They satisfy

$$z_k^{(\ell+1)} = z_k^{(\ell)} + \delta_k \tag{14}$$

where $\delta_k = \boldsymbol{w}_k^T \Delta \boldsymbol{h} > 0$. For the class $i \neq k$ logit,

$$z_i^{(\ell+1)} = z_i^{(\ell)} + \delta_i \tag{15}$$

where $\delta_i = \boldsymbol{w}_i^T \Delta \boldsymbol{h}$. To prove that $\delta_i < 0$, suppose there exist a direction $\boldsymbol{\eta}$ that $\boldsymbol{w}_k^T \boldsymbol{\eta} = 0$ for all $k \in [1, K]$. So we have $\Delta \boldsymbol{h} = \delta_k \boldsymbol{w}_k + \boldsymbol{\eta}$ and,

$$\delta_i = \boldsymbol{w}_i^T \Delta \boldsymbol{h} = \delta_k \boldsymbol{w}_i^T \boldsymbol{w}_k \tag{16}$$

According to $\mathcal{NC}_3$, $\boldsymbol{W}$ form a simplex ETF, meaning all weight vectors have unit norm and the same inner product between any two distinct vectors, i.e., for any $i \neq k$, $\boldsymbol{w}_i^T \boldsymbol{w}_k = \alpha = -\frac{1}{K-1}$. So we have

$$\delta_i = \alpha \delta_k = -\frac{1}{K-1}\delta_k < 0 \tag{17}$$

since $\delta_k > 0$ and $K > 1$.

The softmax output for class $k$ at layers $\ell$ and $\ell + 1$ are given by

$$[\text{SoftMax}(\boldsymbol{z}^{(\ell)})]_k = \frac{e^{z_k^{(\ell)}}}{\sum_{k=1}^K e^{z_k^{(\ell)}}} = \frac{e^{z_k^{(\ell)}}}{e^{z_k^{(\ell)}} + \sum_{i\neq k}^K e^{z_i^{(\ell)}}},$$

$$[\text{SoftMax}(\boldsymbol{z}^{(\ell+1)})]_k = \frac{e^{z_k^{(\ell+1)}}}{\sum_{k=1}^K e^{z_k^{(\ell+1)}}} = \frac{e^{z_k^{(\ell+1)}}}{e^{z_k^{(\ell+1)}} + \sum_{i\neq k}^K e^{z_i^{(\ell+1)}}}$$

$$= \frac{e^{z_k^{(\ell)}+\delta_k}}{e^{z_k^{(\ell)}+\delta_k} + \sum_{i\neq k}^K e^{z_i^{(\ell)}+\delta_i}}$$

For simplify, define $r = \frac{\sum_{i\neq k}^K e^{z_i^{(\ell)}}}{e^{z_i^{(\ell)}}}$, then we have,

$$[\text{SoftMax}(\boldsymbol{z}^{(\ell)})]_k = \frac{1}{1+r}$$

$$[\text{SoftMax}(\boldsymbol{z}^{(\ell+1)})]_k = \frac{e^{\delta_k}}{e^{\delta_k} + re^{\alpha\delta_k}} = \frac{1}{1 + re^{(\alpha-1)\delta_k}}$$

Since $\alpha - 1 = -\frac{1}{K-1} - 1 = -\frac{K}{K-1} < 0$ and $\delta_k > 0$, we have $e^{(\alpha-1)\delta_k} < 1$, and hence

$$[\text{SoftMax}(\boldsymbol{z}^{(\ell+1)})]_k > [\text{SoftMax}(\boldsymbol{z}^{(\ell)})]_k \tag{18}$$

which proves that

$$[\text{SoftMax}(\boldsymbol{W}\boldsymbol{h}_{k,i}^{(\ell+1)})]_k > [\text{SoftMax}(\boldsymbol{W}\boldsymbol{h}_{k,i}^{(\ell)})]_k \tag{19}$$

## B  ADDITIONAL EXPERIMENTS ON VISION MODELS

In this section, we first illustrate the box plots of cosine similarity validate in Appendix B.1. Secondly, we validate that residual connections in transformers resolve feature rotation ambiguity in Appendix B.2. Then, we describe the setup for the aligned training method in Appendix B.3, and demonstrate its effects on transferability in Appendix B.4. Finally, we apply the aligned training methods to the detection transformer in Appendix B.5.

**Setup for Vision Experiments.**  We conduct experiments on both the CIFAR10 and ImageNet1K datasets. The CIFAR10 dataset includes 60,000 color images in 10 classes, each measuring $32 \times 32$ pixels. ImageNet1K contains 1.2 million color images distributed in 1000 classes. To increase the diversity of our training data, we use a data augmentation strategy. This includes random crop and padding, random horizontal flip with a probability of 0.5, and random rotation within 15 degrees. For optimization, we employ AdamW with an initial learning rate of 0.1. This rate decays according to the MultiStepLR at the 100th and 150th epochs, over a total of 200 epochs. We set the weight decay at 1e-4. The global batch size for both datasets is set at 256. For both vision and NLP tasks, we used 4 RTX A5000 GPUs with 24GB of memory each.

### B.1  BOX PLOTS OF SAMPLE-WISE COSINE SIMILARITY

There may be some rare samples with negative sample-wise cosine similarity between features from layers that are far apart.

### B.2  RESIDUAL CONNECTIONS ELIMINATE ROTATION AMBIGUITY

Section 2 demonstrates a consistent trend between COS and CKA. Additionally, when we compute the cosine similarity of features from adjacent layers in Figure 11, most samples exhibit high similarity. These findings suggest that transformers do not have orthogonal transformations across layers. But why does this occur? In this section, we examine the role of skip connections in preventing orthogonal transformations.

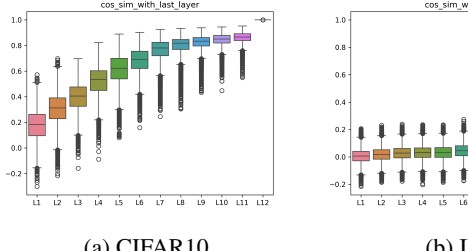

(a) CIFAR10            (b) IN1K

Figure 10: Sample-wise cosine similarity of features from shallow layers and the last-hidden layer. The DeiT-S model is trained with standard training on CIFAR-10 and ImageNet. It shows there are rare samples with negative sample-wise cosine similarity.

Most transformer architectures include skip connections, which are added after the (i) self-attention layer and (ii) MLP layer. According to residual transformation (1), we obtain,

$$\boldsymbol{H}^{(\ell+1)} = \text{MLP}(\text{LN}(\text{MSA}(\text{LN}(\boldsymbol{H}^{(\ell)}) + \boldsymbol{H}^{(\ell)})) + \text{MSA}(\text{LN}(\boldsymbol{H}^{(\ell)})) + \boldsymbol{H}^{(\ell)}$$
$$= \boldsymbol{H}^{(\ell)} + f_{\theta^{(\ell)}}(\boldsymbol{H}^{(\ell)})$$

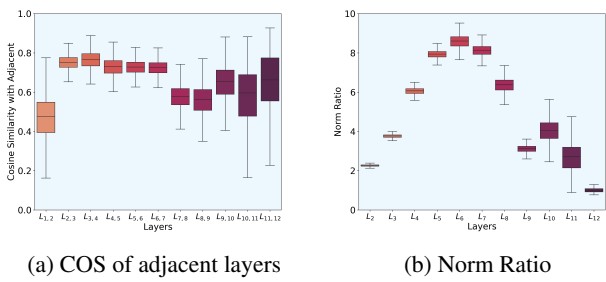

(a) COS of adjacent layers       (b) Norm Ratio

Figure 11: Cosine similarity of features from adjacent layers $\text{COS}(\boldsymbol{h}^{(\ell-1)}, \boldsymbol{h}^{(\ell)})$ and norm ratios $\|\boldsymbol{h}^{(\ell)}\|/\|f_{\theta^{(\ell)}}(\boldsymbol{h}^{(\ell)})\|$ distributions. The DeiT-Small model is trained on Imagenet-1K and evaluated on its validation dataset.

And the feature $\boldsymbol{h}^{(\ell)}$ is a special token of $\boldsymbol{H}^{(\ell)}$. To investigate the effect of residual connections, we calculate the norm ratio $\|\boldsymbol{h}^{(\ell)}\|/\|f_{\theta^{(\ell)}}(\boldsymbol{h}^{(\ell)})\|$, which is the ratio of the norm of skip connection $\boldsymbol{h}^{(\ell)}$ to the norm of the long branch $f_{\theta^{(\ell)}}(\boldsymbol{h}^{(\ell)})$. The results are displayed in Figure 11. High norm ratios suggest that skip connections significantly influence the representational structure of ViT.

To provide further evidence that residual connections resolve the rotation ambiguity, we compared the MLP model with and without these connections and computed their COS and CKA values. For the MLP model without residual connections, as shown in Figure 12(a), the CKA value is not consistent with accuracy and cosine similarity. A high CKA value might indicate significant similarity between features across layers, but it does not necessarily correlate with high classification accuracy. This inconsistency primarily results from the fact that CKA does not account for rotation in the feature space, suggesting that features could rotate without the residual connections. In contrast, for the MLP model with residual connections, as depicted in Figure 12(b), the CKA value aligns with layerwise accuracy, indicating that residual connections effectively eliminate the rotation ambiguity of features.

### B.3   ALIGNED TRAINING DETAILS

**Illustration of Train Once and Fit all devices.** Figure 13 illustrate how aligned training support train once and fit all devices. After aligned training, one can directly fetch from shallow to deep layers of transformer according to the device computational resources and memory constrains.

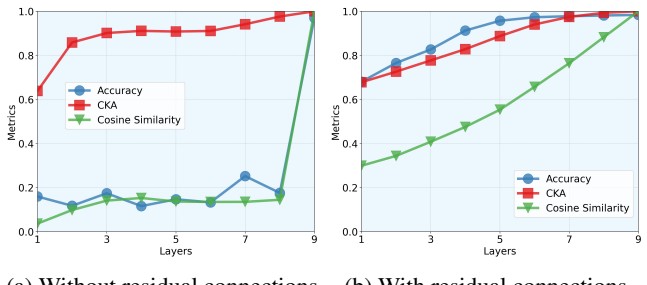

(a) Without residual connections    (b) With residual connections

Figure 12: Comparison of layerwise accuracy, COS(COsine Similarity), and CKA (Centered Kernel Alignment) with the last layer of the 9-layer MLP models with and without residual connection on the MNIST validation dataset. The models are trained from scratch using standard training. In the left figure, CKA fails to accurately reflect the change in layerwise accuracy for the MLP without residual connection. In the right figure, the presence of a residual connection is the reason why CKA works well, as it helps eliminate rotation ambiguity.

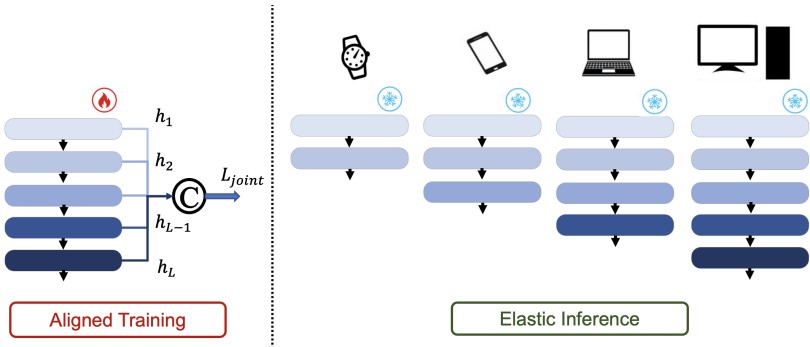

Figure 13: Aligned training of transformer using joint CE loss of all layer features with common classifier and elastic inference for different memory constrains. Once the model is trained using the aligned method, it can fit all devices. Features from darker layers indicate better performance.

**Alternative Approach for Enhancing Layer-wise Representation Similarity.**    In addition to using aligned training loss to enhance similarity, another method is to add the cosine similarity as a regularization term to the loss function.

$$\mathcal{L}_{\text{sim}}(\boldsymbol{h}^{(\ell)}, \boldsymbol{h}^{(L)}) = \sum_{l=1}^{L} \lambda_\ell (1 - \cos(\boldsymbol{h}^{(\ell)}, \boldsymbol{h}^{(L)}))$$

And the total loss is the sum of this two term:

$$\mathcal{L}_{\text{CE-reg}}(\boldsymbol{x}, y) = \mathcal{L}_{\text{CE}}(\boldsymbol{W}\boldsymbol{h}^{(L)}, y) + \beta \mathcal{L}_{\text{sim}}(\boldsymbol{h}^{(\ell)}, \boldsymbol{h}^{(L)})$$

where $\beta > 0$ is the regularization coefficient. According to Figure 14, the regularization term contributes minimally to the improvement of cosine similarity and layer-wise accuracy, compared to aligned training methods.

Using the CE-reg loss results in poor layerwise accuracy and lower cosine similarity compared to the aligned loss. The likely reason for this is an imbalance between the COS alignment objective and the primary classification objective. Our intuition is that directly optimizing for high COS alignment may fail because the COS alignment loss primarily focuses on aligning features across layers, without necessarily making the features discriminative enough for the classification task. In contrast, the cross-entropy (CE) loss directly optimizes for classification, and as a consequence, it naturally improves COS alignment. This suggests that while COS alignment is important, it may not be sufficient on its own without the robust guidance provided by the CE loss.

Another approach involves adding the CKA term as a regularization term to the CE loss. However, this approach may not be effective and has several drawbacks. First, CKA is not always reliable in

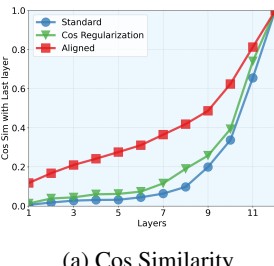
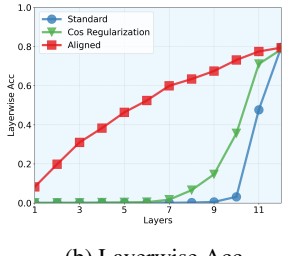

(a) Cos Similarity    (b) Layerwise Acc

Figure 14: Cosine similarity with last layer and layerwise accuracy of standard training using $\mathcal{L}_{\text{CE}}(\boldsymbol{W}\boldsymbol{h}^{(\ell)}, y)$, standard-Reg training using $\mathcal{L}_{\text{CE-reg}}(\boldsymbol{x}, y)$ and Aligned training using $\mathcal{L}_{\text{aligned}}(\boldsymbol{x}, y)$ . The 12 layers DeiT model is trained on ImageNet1K dataset. Regularization term helps little for improving the cosine similarity and layerwise accuracy. But our aligned training improves both a lot.

representing layer-wise similarity across all settings, particularly in scenarios with rotation ambiguity or when residual connections are absent. Second, it is computationally expensive, as it requires computing the Gram matrix to evaluate relationships between features. Lastly, CKA might not perform better than the COS regularization term and may yield similar results, falling short compared to the aligned loss. In transformers, both COS and CKA measure feature similarity and tend to exhibit similar trends. As shown in Figure 14, the COS regularization term contributes minimally to improving cosine similarity and layer-wise accuracy. Based on this, we infer that using CKA as a regularization term would similarly have a minimal impact on enhancing these metrics. Therefore, aligned training approaches may be more effective than relying solely on regularization terms.

**Setup for Aligned Training in ViT.** It's reported (Xin et al., 2021) that training only with this aligned loss would cause the performance drop in the last layer. So following (Xin et al., 2021), we choose the "alternating" training approach, which alternates objectives based on the iteration number. During odd-numbered iterations, we use the CE loss of the final layer $\mathcal{L}_{\text{CE}}(\boldsymbol{W}\boldsymbol{h}_L, y)$. For even-numbered iterations, the strategy involves using the aligned loss $\mathcal{L}_{\text{aligned}}(\boldsymbol{x}, y)$.

Note that this training strategy, which uses a common classifier, no longer requires the KL-divergence term that is commonly used in mutli-exit/classifeirs training. This is because the deep layers have been trained to capture the abstract and discriminative features of the input data, effectively serving as the teacher model. The KL-divergence term is typically used to guide the shallower layers. However, when we use a common classifier, our aligned training method becomes a latent knowledge self-distillation method. The shallow layers can mimic or align their feature representations with those of the deep layers by aligning with the common classifier. As such, the deep layers, with their advanced feature representations, act as the teachers, while the shallow layers, in their quest to improve their feature extraction capabilities, assume the role of students. Therefore, the KL-divergence term is no longer necessary.

**Linear Increasing Weight vs. Uniform Weight for Loss.** In (5), we define the layer weights to increase linearly as $\lambda_\ell = 2\ell/(L(L+1))$. For the ablation study, we also consider a uniform weighting strategy, where all layers are assigned the same weight, i.e., $\lambda_\ell = 1/L$. As shown in Figure 15, using uniform weights in the loss function tends to improve the performance of shallow layers but degrade the final layer. This occurs because shallow layers, which typically learn general but less informative features, produce larger losses, while deeper layers achieve smaller losses. Uniformly weighting all layers disproportionately emphasizes shallow layers and diminishes the importance of deeper ones. In contrast, linearly increasing the weights places greater emphasis on deeper layers, resulting in superior accuracy for the final layer. Therefore, linear increasing weight is selected instead of uniform weight for aligned training.

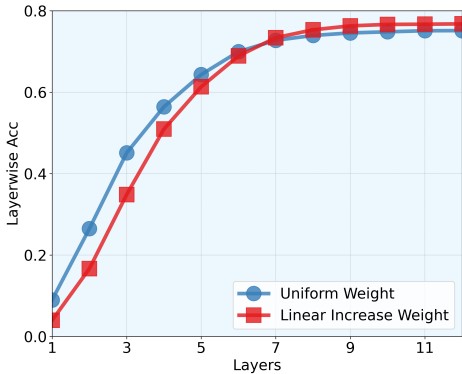

Figure 15: Comparison of linear increasing weight vs. uniform weight for loss function design using Deit-Small on ImageNet-1K.

**Aligned Training Enhance Neural Collapse.** The aligned training method aligns features with the last-layer classifier from shallow to deep layers, enhancing the neural collapse ($\mathcal{NC}$) phenomenon across layers. As shown in Figure 16, aligned training promotes progressive compression as features move closer to the last layer by noting that aligned results in lower $\mathcal{NC}1$ across layers, where intermediate layers increasingly exhibit the stronger $\mathcal{NC}1$ than standard model.[5]

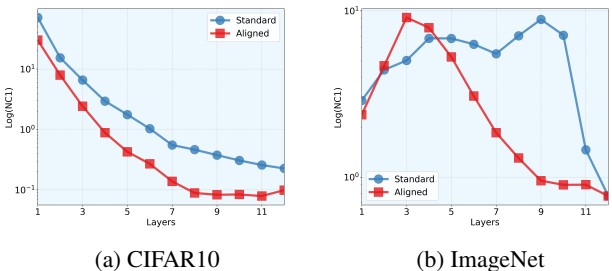

(a) CIFAR10             (b) ImageNet

Figure 16: Comparison of layerwise neural collapse between standard training and aligned training.

## B.4 EFFECTS ON TRANSFER ABILITY

It is often claimed that shallow layers learn universal patterns while deep layers fit to class labels. Questions arise about whether the proposed aligned training approach is that aligning shallow layer features with deep layer features could cause the shallow layers to lose their transfer ability. To resolve this question, we conduct two sets of experiments:

- **Distribution shift**: we first train a DeiT on CIFAR10 with standard training and align training, and then evaluate the layer-wise accuracy on CIFAR10.2 (Lu et al., 2020),
- **Transfer to different tasks**: we first train a DeiT on ImageNet with standard training and align training, and then evaluate the layer-wise accuracy on CIFAR10 by *only* fine-tune a linear classifier, with the feature mapping fixed.

The results are plotted in Figure 17. We observe that for both cases, the distribution shift and transferring to different tasks, layer-wise accuracy curves resemble those on the pre-trained datasets

---

[5]Within-class variability collapse (Papyan et al., 2020; Zhu et al., 2021) for features $\{\boldsymbol{h}_{k,i}\}$ from each layer is computed as $\mathcal{NC}_1 = \frac{1}{K}\operatorname{Tr}(\boldsymbol{\Sigma}_W\boldsymbol{\Sigma}_B^{\dagger})$, where $\boldsymbol{\Sigma}_W = \frac{1}{nK}\sum_{k=1}^{K}\sum_{i=1}^{n}(\mathbf{h}_{k,i} - \overline{\mathbf{h}}_k)(\mathbf{h}_{k,i} - \overline{\mathbf{h}}_k)^{\top}$ captures the within-class covariance, $\boldsymbol{\Sigma}_B = \frac{1}{K}\sum_{k=1}^{K}(\overline{\mathbf{h}}_k - \overline{\boldsymbol{h}})(\overline{\mathbf{h}}_k - \overline{\boldsymbol{h}})^{\top}$ represents the between-class covariance, $\overline{\boldsymbol{h}}_k$ represents the class-mean features, and $\overline{\boldsymbol{h}}$ represents the global mean of the features.

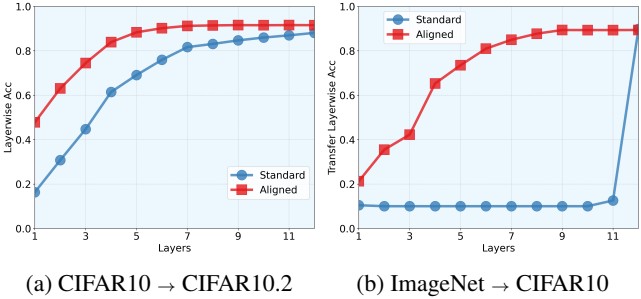

(a) CIFAR10 → CIFAR10.2          (b) ImageNet → CIFAR10

Figure 17: The comparison of layer-wise accuracy between a standard model and an aligned model.

shown in Figure 4 and Figure 5, demonstrating that aligned training not only improves layer-wise accuracy for the pre-trained datasets but also for the downstream datasets. In other words, the aligned training methods maintain transferability, ensuring that the trained model can be effectively transferred.

## B.5 ALIGNED TRAINING FOR DETECTION TRANSFORMER

In this section, we demonstrate that aligned training is not only beneficial for classification tasks but also effective for other tasks such as object detection. Following the DeTr framework (Carion et al., 2020), which employs an encoder-decoder architecture, the encoder extracts global image features, while the decoder predicts object classes and their bounding boxes using queries. Aligned training can be applied to the decoder, where predictions are made using intermediate features from its layers. This is achieved through auxiliary decoding losses (Carion et al., 2020). We evaluate the AP50 (average precision at 50% IoU) for predictions exiting from each decoding layer using aligned training and compare it with predictions from the last layer of standard training, as shown in Figure 18. As noted in DeTr (Carion et al., 2020), the inclusion of aligned training losses is critical for performance, and removing them results in a significant drop in accuracy.

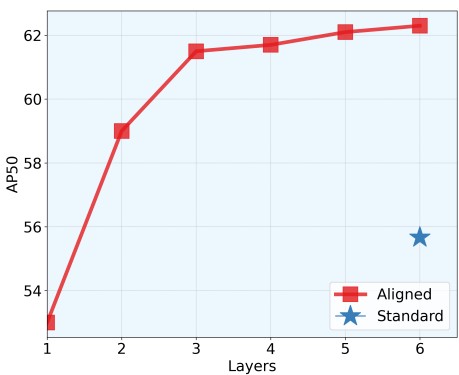

Figure 18: Comparison of aligned training (using auxiliary decoding losses) and standard training ( using last layer loss only) of DeTr model.

## C ADDITIONAL EXPERIMENTS ON LANGUAGE MODELS

In this section, we first validate that the saturation events described in Section 2.2 are consistently observed in LLaMA3, as shown in Appendix C.1. Then we show aligned training on BERT and GPT models in Appendix C.2.

## C.1 Saturate Events in Llama3

In this section, we verify that saturation events, as described in Section 2.2, also occur in large language models such as LLaMA3 (Dubey et al., 2024). As shown in Figure 19, we compare the saturation events across two variants of the LLaMA3 model: the 24-layer LLaMA3.2 3B model and the 32-layer LLaMA3.1 8B model. Both models are given the same prompt to ensure a controlled comparison. The figure demonstrates how saturation events emerge across layers in models of varying sizes.

Furthermore, as shown in Figure 20, we examine a specific instance of saturation events in the 28-layer LLaMA3.2 3B model. Using the prompt "Simply put, the theory of relativity states," the model generates predictions over 24 decoding steps with greedy decoding. The figure displays the predicted words at each layer, using the last-layer classifier to obtain outputs. The color gradient represents the softmax logits of the last-layer predictions, with red indicating a low probability (0) and blue indicating a high probability (1). Saturation events are observed when the prediction at a given layer $\ell$ remains unchanged through subsequent layers until the final output. This visualization highlights the occurrence of saturation events during the progression of intermediate representations and their impact on the model's final predictions.

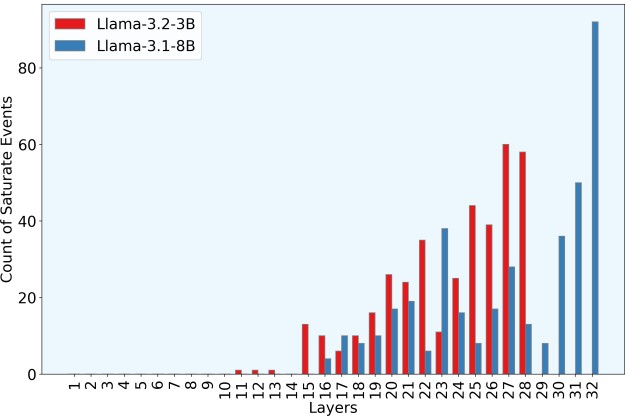

Figure 19: Saturation events for a 24-layer LLaMA3.2 3B model and a 32-layer LLaMA3.1 8B model using the same input prompt.

## C.2 Aligned Training on Language Models

In this section, we provide more details about the datasets and computational resources used. The datasets (GLUE, and Wikitext-103) are publicly available under the MIT license. For NLP tasks, we used a single RTX A5000 GPUs with 24GB of memory. Then, we will introduce the experimental setup for AlignedBERT and AlignedGPT models.

**Setup in AlignedBERT**   The General Language Understanding Evaluation (GLUE) benchmark comprises nine tasks for assessing natural language understanding. In our AlignedBERT experiments on the GLUE dataset, we used a sequence length of 256. We employed AdamW for optimization with an initial learning rate of 2e-5, and a batch size of 32. Each task underwent fine-tuning for three epochs. The WikiText-103 language modeling dataset consists of over 100 million tokens extracted from Wikipedia's verified good and featured articles. For AlignedGPT experiments on the WikiText-103 dataset, we maintained the sequence length at 256 and used AdamW with an initial learning rate of 2e-5. In this case, we set the batch size to 8.

**Setup in AlignedGPT.**   Our aligned training method can be used with any transformer-based language models. In this study, we evaluated our method using the GPT-2 model. We finetune the GPT2 models using aligned training methods and then use intermediate layers of GPT2 to generate the texts.

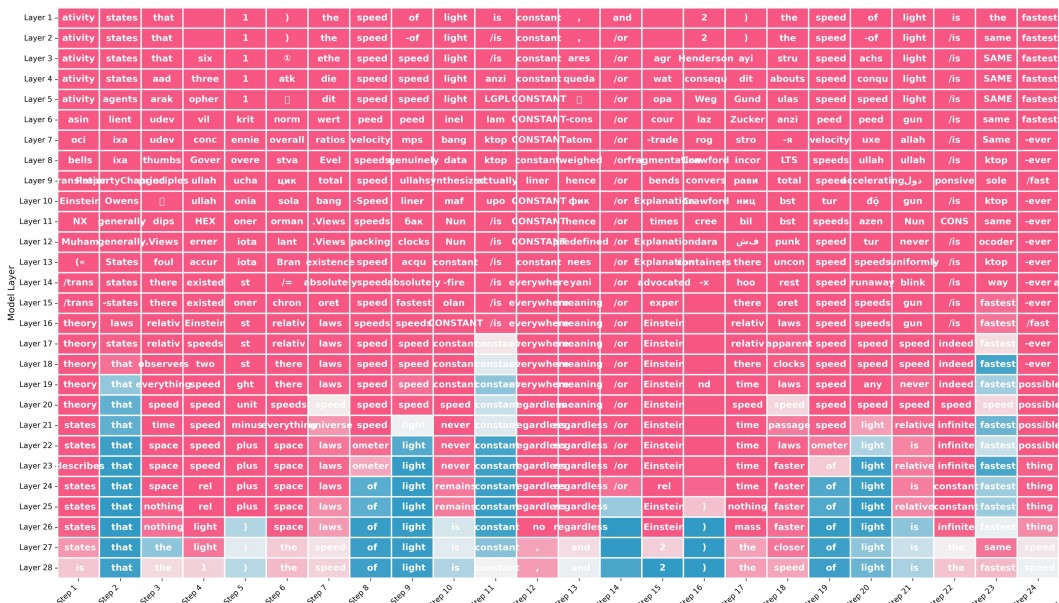

Figure 20: Specific instance of saturation events for LLaMA3.2 3B model responding to the prompt: "Simply put, the theory of relativity states " using greedy decoding. Predicted words are shown for each layer, with saturation events identified where predictions at layer $\ell$ remain unchanged until the final layer.

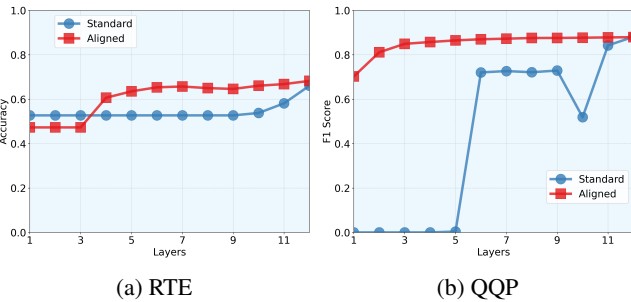

(a) RTE      (b) QQP

Figure 21: Layerwise accuracy for **AlignedBERT** and BERT and on RTE and QQP datasets of the GLUE benchmark.

- **Model and Baselines** We finetune GPT-2 on the Wikitext-103 dataset with the proposed objective $\mathcal{L}_{\text{aligned}}$ for $40k$ training steps and generate the text continuation with nucleus sampling (Holtzman et al., 2019) with $p = 0.95$ decoding methods. For the standard baseline model, we finetune the model with CE loss $\mathcal{L}_{\text{MLE}}$. The model is finetuned using a single 24G RTX A5000 GPU for 70 hours.

- **Evaluations** Following (Su & Collier, 2023), we evaluate the model from two perspectives: (1) language modeling quality, assessing the inherent quality of the model, and (2) generation quality, measuring the quality of the text the model produces. In assessing language modeling quality, we calculate the prediction accuracy and perplexity of each layer. When evaluating generation quality, we measure the similarity between the prompt text and generated text using coherence. We employ generation repetition to gauge the diversity of the generated text. The metrics are defined as follows,

  - **Prediction Accuracy** The accuracy is computed on the Wikitext-103 test set as,

$$\textbf{Acc} = \frac{1}{D} \sum_{i=1}^{D} \sum_{i=1}^{n} \mathbb{1}[\arg\max p_\theta(x|\boldsymbol{x}_{<i}) = x_i] \tag{20}$$

where the $D$ is the number of samples in the test dataset.

– **Perplexity** The perplexity is computed on the test set of Wikitext-103. It's computed as the exponential of the test loss.

– **Coherence** Coherence measures the relevance between the prefix text and the generated text. We apply the advanced sentence embedding method, SimCSE (Gao et al., 2021), to measure the semantic coherence or consistency between the prefix and the generated text. The coherence score is defined as follows,

$$\textbf{Coherence} = \boldsymbol{h}_{\boldsymbol{x}}^T \boldsymbol{h}_{\hat{\boldsymbol{x}}} / \|\boldsymbol{h}_{\boldsymbol{x}}\| \|\boldsymbol{h}_{\hat{\boldsymbol{x}}}\| \tag{21}$$

where $\boldsymbol{x}$ is the prefix text and $\hat{\boldsymbol{x}}$ is the generated text and $\boldsymbol{h}_{\boldsymbol{x}} = \text{SimCSE}(\boldsymbol{x})$ and $\boldsymbol{h}_{\hat{\boldsymbol{x}}} = \text{SimCSE}(\hat{\boldsymbol{x}})$. Higher coherence means more correlation to the given prompt.

– **Diversity** Diversity measures the occurrence of generation at different n-gram levels. It is defined as:

$$\textbf{Diversity} = \prod_{n=2}^{4} \frac{|\text{unique n-grams}(\hat{\boldsymbol{x}})|}{|\text{total n-grams}(\hat{\boldsymbol{x}})|} \tag{22}$$

A higher diversity score suggests fewer repeated words in the generated text.

## D  ADDITIONAL EXPERIMENTS ON MULTI-MODALITY MODELS

In this section, we demonstrate that the observation in Section 2.1 can also extend to multi-modality models such as the pre-trained CLIP models (Radford et al., 2021). Given that the CLIP model comprises both a vision encoder and a text encoder, we evaluate the cosine similarity within each modality (vision or text) and across modalities (between the vision encoder and text encoder). This comprehensive analysis highlights the robustness of the observed phenomena across diverse components of the CLIP architecture.

Specifically, given a pretrained CLIP model with vision encoder $f_{\text{vision}}$ and text encoder $f_{\text{text}}$, we extract the $\ell$-th layer vision feature $\hat{\boldsymbol{v}}^{(\ell)} \in \mathbb{R}^{768}$ by taking the corresponding [CLS] token outputs of $\ell$-th layer from $f_{\text{vision}}$; similarly, take the $\ell$-th layer vision feature $\hat{\boldsymbol{t}}^{(\ell)} \in \mathbb{R}^{512}$ from the [EOS] token outputs of $f_{\text{text}}$. Since CLIP projects both the vision features and text features to the same embedding space through a vision projection matrix $\boldsymbol{W}_{\text{vision}} \in \mathbb{R}^{768 \times 512}$ (followed by a layer normalization LN) and a text projection matrix $\boldsymbol{W}_{\text{text}} \in \mathbb{R}^{512 \times 512}$ (also followed by a layer normalization LN), we also apply these projection matrices to the hidden layer features to obtain

$$\boldsymbol{v}^{(\ell)} = \text{LN}(\boldsymbol{W}_{\text{vision}}\hat{\boldsymbol{v}}^{(\ell)}) \in \mathbb{R}^{512}, \quad \boldsymbol{t}^{(\ell)} = \text{LN}(\boldsymbol{W}_{\text{text}}\hat{\boldsymbol{t}}^{(\ell)}) \in \mathbb{R}^{512}. \tag{23}$$

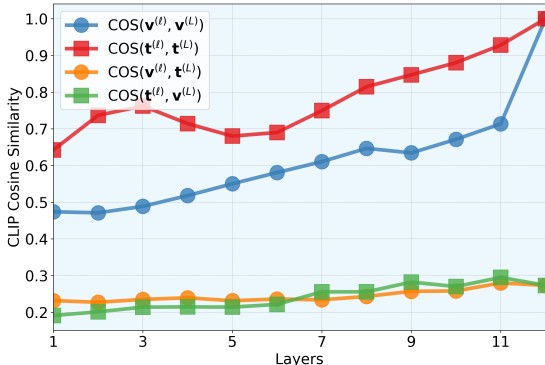

Figure 22: Illustration of cosine similarity between the hidden layer to the last layer for within and cross modalities in a 12-layer CLIP-B/32.

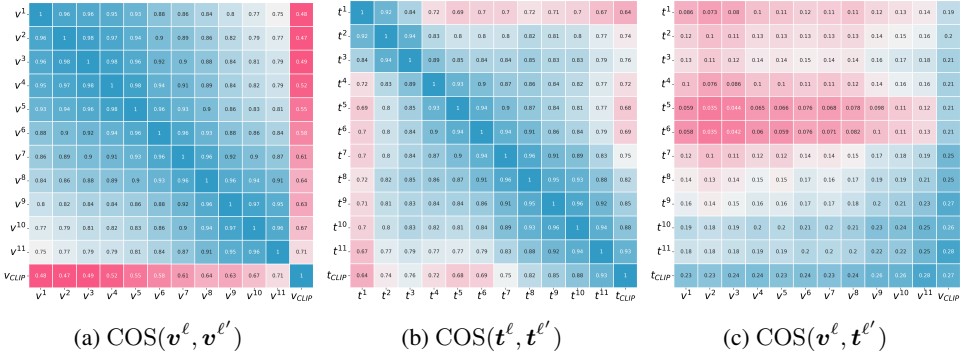

$$\text{(a) } \mathrm{COS}(\boldsymbol{v}^\ell, \boldsymbol{v}^{\ell'}) \qquad \text{(b) } \mathrm{COS}(\boldsymbol{t}^\ell, \boldsymbol{t}^{\ell'}) \qquad \text{(c) } \mathrm{COS}(\boldsymbol{v}^\ell, \boldsymbol{t}^{\ell'})$$

Figure 23: Layer-wise cosine similarities within (a) vision encoder, (b) text encoder, and (c) cross vision and text encoder for a pretrained 12-layers CLIP-B/32.

To measure within-modality cosine similarity, we calculate layer-wise similarity within the vision encoder as $\mathrm{COS}(\boldsymbol{v}^{(\ell)}, \boldsymbol{v}^{(\ell')})$ and within the text encoder as $\mathrm{COS}(\boldsymbol{t}^{(\ell)}, \boldsymbol{t}^{(\ell')})$. Similarly, for layer-wise cross-modality cosine similarity, we evaluate the relationships between the vision encoder and text encoder using $\mathrm{COS}(\boldsymbol{v}^{(\ell)}, \boldsymbol{t}^{(\ell')})$ and $\mathrm{COS}(\boldsymbol{t}^{(\ell)}, \boldsymbol{v}^{(\ell')})$. Figure 22 plots the layer-wise within-modality similarity and cross-modality similarity by comparing the hidden-layer features to the last-layer features, while Figure 23 plots all the pair-wise results. All experiments are conducted on the CIFAR10 validation dataset, where the text input to the text encoder is: "This is a photo of a {label}". From both figures, we observe a clear pattern of progressively increasing layer-wise representational similarity in both the vision encoder and the text encoder. For cross-modality similarity, we note that the last-layer vision and text representations are not perfectly aligned (e.g., $\mathrm{COS}(\boldsymbol{t}^{(L)}, \boldsymbol{v}^{(L)})$ is not close to 1), a phenomenon commonly referred to as the *modality gap*, which has been consistently observed across various multi-modal models (Liang et al., 2022). Despite this, we also observe a progressive increase in layer-wise representation similarity across modalities. Together, these results highlight a distinct trend of progressively increasing layer-wise representational similarity within and across modalities.

