$\boldsymbol{h}^{(L)}$ to make predications as[1] $g(\boldsymbol{h}^{(L)}) = \arg\max_j [\text{SoftMax}(\boldsymbol{W}\boldsymbol{h}^{(L)})]_j$, where $[\cdot]_j$ denotes the $j$-th

---

[1]There is also a bias term $\boldsymbol{b}$ in classification layer, but we omit it for simplicity of presentation.

entry and $\boldsymbol{W} \in \mathbb{R}^{K \times d}$ maps the $d$ dimensional features to $K$ dimensional logits. We may also directly apply the last layer classifier $g(\cdot)$ to the hidden layer features $\boldsymbol{h}^{(\ell)}$ to make predictions via $g(\boldsymbol{h}^{(\ell)}) = \arg \max_j [\text{SoftMax}(\boldsymbol{W} \boldsymbol{h}^{(\ell)})]_j$. Given data samples $\mathcal{S} := \{\boldsymbol{x}_{k,i}\}$, where $\boldsymbol{x}_{k,i}$ represents the $i$-th sample of class $k$ with $i \in [n] := \{1, \ldots, n\}$ and $k \in [K]$, we define layer-wise accuracy as

$$\text{Acc}_{\mathcal{S}}^{(\ell)} := \frac{1}{Kn} \sum_{k=1}^{K} \sum_{i=1}^{n} \mathbb{1}[g(\boldsymbol{h}_{k,i}^{(\ell)}) = k]. \tag{2}$$

**Existing work on measuring representational similarity** Similarity analysis is widely applied in the literature, including research on learning dynamics (Morcos et al., 2018; Mehrer et al., 2018), effects of width and depth (Nguyen et al., 2020), differences between supervised and unsupervised models (Gwilliam & Shrivastava, 2022), robustness (Jones et al., 2022; Nanda et al., 2022), evaluating knowledge distillation (Stanton et al., 2021), language representation (Kudugunta et al., 2019; Shi et al., 2022), and generalizability (McCoy et al., 2019; Lee et al., 2022; Pagliardini et al., 2022). To enable measuring the similarity of features from from different architectures or layers that have different dimension, most existing methods for analyzing the similarity between representations of high dimensions, such as those based on Canonical Correlation Analysis (CCA) and widely used Centered Kernel Alignment (CKA) (Kornblith et al., 2019) , rely on statistical properties of the representations for a set of data points. For instance, given input feature sequence $\boldsymbol{Z}^\ell = \begin{bmatrix} \boldsymbol{h}_{1,1}^{(\ell)} & \cdots & \boldsymbol{h}_{K,n}^{(\ell)} \end{bmatrix} \in \mathbb{R}^{d \times N}$ denoting the features of $N = Kn$ training samples, the widely-used CKA with a linear kernel quantifies similarities between features $\boldsymbol{Z}^\ell$ and $\boldsymbol{Z}^{\ell'}$ as

$$\text{CKA} = \text{Tr}((\boldsymbol{Z}^{\ell'})^\top \boldsymbol{Z}^{\ell'} \cdot (\boldsymbol{Z}^\ell)^\top \boldsymbol{Z}^\ell) / (\| \boldsymbol{Z}^\ell (\boldsymbol{Z}^\ell)^\top \|_F \| \boldsymbol{Z}^{\ell'} (\boldsymbol{

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

where $x$ is the prefix text and $\hat{x}$ is the generated text and $h_x = \text{SimCSE}(x)$ and $h_{\hat{x}} = \text{SimCSE}(\hat{x})$. Higher coherence means more correlation to the given prompt.

– **Diversity** Diversity measures the occurrence of generation at different n-gram levels. It is defined as:

$$\textbf{Diversity} = \prod_{n=2}^{4} \frac{|\text{unique n-grams}(\hat{x})|}{|\text{total n-grams}(\hat{x})|} \tag{22}$$

A higher diversity score suggests fewer repeated words in the generated text.

## D   ADDITIONAL EXPERIMENTS ON MULTI-MODALITY MODELS

In this section, we demonstrate that the observation in Section 2.1 can also extend to multi-modality models such as the pre-trained CLIP models (Radford et al., 2021). Given that the CLIP model comprises both a vision encoder and a text encoder, we evaluate the cosine similarity within each modality (vision or text) and across modalities (between the vision encoder and text encoder). This comprehensive analysis highlights the robustness of the observed phenomena across diverse components of the CLIP architecture.

Specifically, given a pretrained CLIP model with vision encoder $f_{\text{vision}}$ and text encoder $f_{\text{text}}$, we extract the $\ell$-th layer vision feature $\hat{v}^{(\ell)} \in \mathbb{R}^{768}$ by taking the corresponding [CLS] token outputs of $\ell$-th layer from $f_{\text{vision}}$; similarly, take the $\ell$-th layer vision feature $\hat{t}^{(\ell)} \in \mathbb{R}^{512}$ from the [EOS] token outputs of $f_{\text{text}}$. Since CLIP projects both the vision features and text features to the same embedding space through a vision projection matrix $W_{\text{vision}} \in \mathbb{R}^{768 \times 512}$ (followed by a layer normalization LN) and a text projection matrix $W_{\text{text}} \in \mathbb{R}^{512 \times 512}$ (also followed by a layer normalization LN), we also apply these projection matrices to the hidden layer features to obtain

$$v^{(\ell)} = \text{LN}(W_{\text{vision}}\hat{v}^{(\ell)}) \in \mathbb{R}^{512}, \quad t^{(\ell)} = \text{LN}(W_{\text{text}}\hat{t}^{(\ell)}) \in \mathbb{R}^{512}. \tag{23}$$

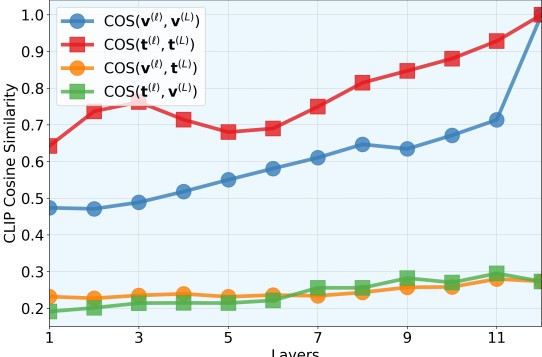

Figure 22: Illustration of cosine similarity between the hidden layer to the last layer for within and cross modalities in a 12-layer CLIP-B/32.

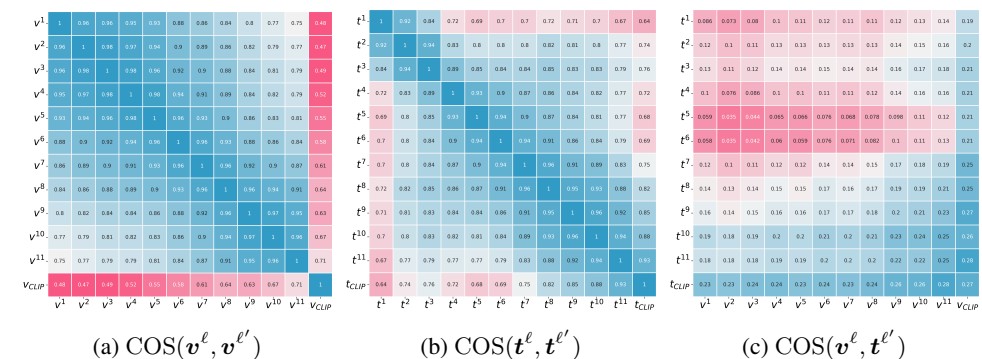

$$\text{(a) COS}(\boldsymbol{v}^\ell, \boldsymbol{v}^{\ell'}) \qquad \text{(b) COS}(\boldsymbol{t}^\ell, \boldsymbol{t}^{\ell'}) \qquad \text{(c) COS}(\boldsymbol{v}^\ell, \boldsymbol{t}^{\ell'})$$

Figure 23: Layer-wise cosine similarities within (a) vision encoder, (b) text encoder, and (c) cross vision and text encoder for a pretrained 12-layers CLIP-B/32.

To measure within-modality cosine similarity, we calculate layer-wise similarity within the vision encoder as $\text{COS}(\boldsymbol{v}^{(\ell)}, \boldsymbol{v}^{(\ell')})$ and within the text encoder as $\text{COS}(\boldsymbol{t}^{(\ell)}, \boldsymbol{t}^{(\ell')})$. Similarly, for layer-wise cross-modality cosine similarity, we evaluate the relationships between the vision encoder and text encoder using $\text{COS}(\boldsymbol{v}^{(\ell)}, \boldsymbol{t}^{(\ell')})$ and $\text{COS}(\boldsymbol{t}^{(\ell)}, \boldsymbol{v}^{(\ell')})$. Figure 22 plots the layer-wise within-modality similarity and cross-modality similarity by comparing the hidden-layer features to the last-layer features, while Figure 23 plots all the pair-wise results. All experiments are conducted on the CIFAR10 validation dataset, where the text input to the text encoder is: "This is a photo of a {label}". From both figures, we observe a clear pattern of progressively increasing layer-wise representational similarity in both the vision encoder and the text encoder. For cross-modality similarity, we note that the last-layer vision and text representations are not perfectly aligned (e.g., $\text{COS}(\boldsymbol{t}^{(L)}, \boldsymbol{v}^{(L)})$ is not close to 1), a phenomenon commonly referred to as the *modality gap*, which has been consistently observed across various multi-modal models (Liang et al., 2022). Despite this, we also observe a progressive increase in layer-wise representation similarity across modalities. Together, these results highlight a distinct trend of progressively increasing layer-wise representational similarity within and across modalities.