# OpenReview forum: "Tracing Representation Progression: Analyzing and Enhancing Layer-Wise Similarity"
_ICLR.cc/2025/Conference — ICLR 2025 Poster_

### Official Review · Reviewer_s3vt · 2024-11-03

**Soundness:** 3
**Presentation:** 3
**Contribution:** 3
**Rating:** 6
**Confidence:** 5

**Summary:**

This paper studied the similarity of representations between the hidden layers of individual transformers and found that a simple cosine similarity metric can be used for similarity evaluation. Experimental results revealed that representations across layers are positively correlated and the authors introduce a multi-exit mechanism. The innovation lies in using the same classifier for different layers.

**Strengths:**

1. The deduction and experiments in the paper are relatively solid. The authors have made efforts to investigate the similarity of representations across different layers in transformers.

2. The paper is well-written and easy to follow.

**Weaknesses:**

Using the same classifier for multi-exit is quite straightforward and its technical contribution seems limited. The experimental results seem not very promising compared with previous multi-exit methods. The authors may need to further enhance the novelty or provide more convincing evidence of the superiority of their approach compared with multi-exit/classifier to make the paper more acceptable.

**Questions:**

1. Why use linearly increasing weights for aligned losses? Experiments for different choices on weights are preferred.

---

> ### Author Response · Authors · 2024-11-25
>
> > Using the same classifier for multi-exit is quite straightforward and its technical contribution seems limited. The experimental results seem not very promising compared with previous multi-exit methods. The authors may need to further enhance the novelty or provide more convincing evidence of the superiority of their approach compared with multi-exit/classifier to make the paper more acceptable.
>
> Thank you for the comment.  We first want to highlight the contribution of this paper. As presented in the abstract and introduction, the main contribution of this paper includes (1) demonstrating the sample-wise cosine similarity as an efficient approach for measuring representation similarity, (2) providing a theoretical justification for the progressively increasing layer-wise representation similarity, offering an explanation for the saturation events, (3) aligned training for enhancing layer-wise representation similarity, leading to multi-exit models with a single classifier. Our work is the first to show that one common classifier is sufficient for multi-exit models. As shown in Table 1, a single classifier can significantly reduce the number of parameters and the computational complexity for multi-exit models, particularly for recent large models with a large number of classes (vocabulary size) and large feature dimensions. If the reviewer has any detailed comments or additional questions, we would be happy to discuss them further.
>
> > Why use linearly increasing weights for aligned losses? Experiments for different choices on weights are preferred.
>
> Thanks for the question. We have added an ablation study on the weight choice for the aligned losses in Appendix Figure 15, comparing the linear increasing weight and uniform weight. Using uniform weights in the loss function tends to improve the performance of shallow layers but degrade the final layer. This occurs because shallow layers, which typically learn less informative features, produce larger losses, while deeper layers achieve smaller losses. Thus, uniformly weighting all layers disproportionately emphasizes shallow layers and diminishes the importance of deeper ones. In contrast, linearly increasing the weights places greater emphasis on deeper layers, resulting in superior accuracy for the final layer. Therefore, linear increasing weight is selected instead of uniform weight for loss design. We have incorporated this discussion in the Appendix.

---

> > ### Comment · Reviewer_s3vt · 2024-11-28
> >
> > Thank the authors for their response, which has addressed my concerns. updated my score to 6.

---

> > > ### Author Response · Authors · 2024-11-28
> > >
> > > Thank you for reading our rebuttal and raising score. We truly appreciate your recognition of our work.

---

### Official Review · Reviewer_aKPD · 2024-11-03

**Soundness:** 3
**Presentation:** 3
**Contribution:** 3
**Rating:** 6
**Confidence:** 2

**Summary:**

This paper focuses on the analysis of layer-wise representations of Transformers. They demonstrates a series of beneficial observations, e.g., representations across layers are positively correlated. Meanwhile, they find the model's top prediction remains unchanged across subsequent layers. Following the observation, they further propose an aligned training strategy to improve the effectiveness of shallow layer, which is able to provide more early saturation events, minimal depth needed for the given task, multi-exit models.

**Strengths:**

1. By analyzing the similarity of representations among different layers, the paper demonstrates the possibility of early saturation events and shared classifier among different layers.
2. They further presents a training strategy to improve effectiveness of shallow layers, such that they can enjoy more early saturation events, minimal depth, and so on
3. Some analysis in this paper is insightful, e.g., the shadow layer is able to achieve approaching performance with the depth layer. This might inspire more efficient large model inference.
4. The experiments are comprehensive.

**Weaknesses:**

1. a small weakness is that the previous approaches have observed this phenomenon that representations in the early layers can also achieve reasonable classifiers, though I think this is a tiny issue.
Please refer to the Questions.

**Questions:**

I have a few questions about this submission.
1. According to Figure 6, does the proposed aligned training strategy decrease the performance more clearly in the last layers in large models? Please discuss the performance trade-offs between shallow and deep layers across different model sizes.
2. With the common classifier, is there still a neural collapse phenomenon in the model? Is there any difference of this phenomenon across different layers? I'd like to see some discussions about this.

---

> ### Author Response · Authors · 2024-11-25
>
> > According to Figure 6, does the proposed aligned training strategy decrease the performance more clearly in the last layers in large models? Please discuss the performance trade-offs between shallow and deep layers across different model sizes.
>
> Thank you for your valuable and constructive feedback. Figure 6 is not about the performance, the lines represent the cosine similarity of each layer compared to the final layer, and bars indicate the number of saturate events occurring at each layer. Across all cases (a-c), models with aligned training show more samples exiting at earlier layers compared to standard training, indicating improved model efficiency. When comparing cases (a) and (b), we observe that ImageNet samples tend to exit at later layers than CIFAR10 samples, which is expected due to the greater complexity of ImageNet. If we misunderstood the reviewer’s comments, we would appreciate if the reviewer could further elaborate on the comments, which we will be happy to address.
>
> > With the common classifier, is there still a neural collapse phenomenon in the model? Is there any difference of this phenomenon across different layers? I'd like to see some discussions about this.
>
> Thank you very much for the question. We present additional results on the impact of aligned training on neural collapse in Appendix B.3. Our experiments demonstrate that aligned training generally results in greater layer-wise neural collapse. This occurs because aligned training promotes layer-wise representation similarity, thereby encouraging progressive compression and separation from shallow to deep layers.

---

> > ### Comment · Reviewer_aKPD · 2024-11-30
> >
> > Thanks for your reply. My concern has been marginally relieved.

---

> > > ### Author Response · Authors · 2024-11-30
> > >
> > > Thank you for reading our rebuttal. We truly appreciate your recognition of our work.

---

### Official Review · Reviewer_xxse · 2024-11-04

**Soundness:** 3
**Presentation:** 3
**Contribution:** 3
**Rating:** 8
**Confidence:** 3

**Summary:**

This paper focuses on understanding the behavior of deep neural networks, particularly transformer models, by examining the similarity of internal representations across hidden layers. The authors introduce a sample-wise cosine similarity metric that aligns with more complex statistical methods and reveals increasing representation similarity as layers get closer. They provide a theoretical justification for this phenomenon under the geodesic curve assumption and demonstrate that enhanced representation similarity leads to increased predicted probability and earlier saturation events in model predictions. The paper proposes an aligned training method to improve shallow layer effectiveness, resulting in more early saturation events and higher layer-wise accuracies. Finally, the authors show that their approach enables multi-exit models with a single classifier, reducing parameter count and computational complexity while maintaining performance.

**Strengths:**

In general, the proposed paper is well-written, the author provides detailed experiments, extensive theoretical analysis to analysis the feature pattern in Transformers. Based on these analyses, the author proposes a aligned training method for enhancing shallow layer performance. The proposed method achieves performance gain and speed boost on CV and NLP tasks.

**Weaknesses:**

In general, I think the proposed paper is well formulated and written. However, I still have some concerns about the paper:

1. I don't think it is useful for enhancing similarity between all Transformer outputs (representations), it may help in simple tasks like image classification and sentence classification. But for complex tasks (like object detection/semantic segmentations), we may not want all representations to be similar. I think similar tasks may exist in NLP tasks (like parsing), Then the author should talk about the limitations or give more experiments to verify the effectiveness of the proposed method.

2. In the paper, the author performs experiments on small datasets and small models, like DeiT trained on CIFAR10 and ImageNet. Also Bert/GPT2 on small NLP tasks. If the data increases, like a pre-trained CLIP/OpenCLIP on large datasets, will the findings/analyses be the same? Moreover, if the model becomes larger (like a LLM like Llama3), will the findings/analyses be the same? Does the saturation events still occur in those models? I'm curious about that.

**Questions:**

Based on the weaknesses, I have some questions.

1. Is enhancing similarity beneficial for all tasks besides the simple classification task?

2. When enlarging the data/model size, will the method/analyses still works?

---

> ### Author Response · Authors · 2024-11-25
>
> > Is enhancing similarity beneficial for all tasks besides the simple classification task? I don't think it is useful for enhancing similarity between all Transformer outputs (representations), it may help in simple tasks like image classification and sentence classification. But for complex tasks (like object detection/semantic segmentations), we may not want all representations to be similar. I think similar tasks may exist in NLP tasks (like parsing), Then the author should talk about the limitations or give more experiments to verify the effectiveness of the proposed method.
>
> Thank you very much for your valuable and constructive feedback. First, note that we have verified the performance of aligned training methods not only for image classification and sentence classification but also for text generation, which is now often regarded as a multi-task learning paradigm for NLP tasks. That said, following the reviewer’s suggestion, we conducted additional experiments on object detection with a detection transformer (DeTr). The results in Appendix B.5 show that by aligned training, the decoders' shallow layers of DeTr can also effectively predict both class and bounding box, even achieving better AP50 scores than the final layer performance of standard training. Interestingly, we observed that a similar approach to aligned training, called auxiliary decoding losses, is employed in DeTr. Our analysis provides a justification for this improved performance, as it enhances token feature similarity across layers.
>
> > When enlarging the data/model size, will the method/analyses still works? In the paper, the author performs experiments on small datasets and small models, like DeiT trained on CIFAR10 and ImageNet. Also Bert/GPT2 on small NLP tasks. If the data increases, like a pre-trained CLIP/OpenCLIP on large datasets, will the findings/analyses be the same? Moreover, if the model becomes larger (like a LLM like Llama3), will the findings/analyses be the same? Does the saturation events still occur in those models? I'm curious about that.
>
> Thank you for your valuable and constructive feedback. Following the reviewer’s suggestion, to validate the scalability of our findings, we conducted additional experiments on multi-modality models (the CLIP model) and advanced LLM (Llama3). The additional results in Appendix D (Figures 22 and 23) demonstrate that the sample-wise, layer-wise representational similarity observed in Section 2.1 extends to CLIP models. Notably, we observe a progressive increase in layer-wise representation similarity not only within each modality (vision or text), but also across modalities (between vision and text encoders). The additional results in Appendix C.1 (Figures 19 and 20) show that  the saturation events described in Section 2.2 are also present in both Llama3.2 3B (28 layers) and Llama3.1 8B (32 layers).

---

> > ### Comment · Reviewer_xxse · 2024-11-27
> > **Response to the author**
> >
> > Thanks the author for the response. I think it has addressed my concern, then I slightly increase my score to 8.

---

> > > ### Author Response · Authors · 2024-11-27
> > >
> > > Thank you for reading our rebuttal and raising score. We truly appreciate your recognition of our work.

---

### Official Review · Reviewer_ABue · 2024-11-09

**Soundness:** 2
**Presentation:** 3
**Contribution:** 2
**Rating:** 6
**Confidence:** 3

**Summary:**

This work studies the feature similarity of neural networks and reveals that: (I) simple-wise cosine similarity can capture representation similarity; (ii) saturation events related with feature similarity and based on this observations, this work proposed a aligned training approach to enhance the representation thus benefit the performance and also the multi-exit inference approach.

**Strengths:**

- The manuscript is logically organized, including the observations and its applications.
- Both empirical and theoretical justifications are provided.
- The study covers both the vision and language domains.

**Weaknesses:**

- Line 342 “To the best of our knowledge, our work is the first to show that one common classifier is sufficient for multi-exit models.” This is not true, a lot of early exiting methods can do with a single classifier heads [1-3].
- Line 240: “Progressively increasing layer-wise representation similarity”, thus observations might be different in other domain[2], is there any insights why autoregressive models seems not have progressively increasing layer-wise representation similarity?
- Missing previous literature[4] which also studies layer-wise cosine similarity, yet in the language domain.

[1] https://proceedings.neurips.cc/paper_files/paper/2022/file/6fac9e316a4ae75ea244ddcef1982c71-Paper-Conference.pdf

[2] https://arxiv.org/pdf/2404.03865

[3] https://arxiv.org/pdf/2403.03853

[4] https://arxiv.org/pdf/2202.08625

**Questions:**

Please refer to the weaknesses part.

---

> ### Author Response · Authors · 2024-11-25
>
> > Line 342 “To the best of our knowledge, our work is the first to show that one common classifier is sufficient for multi-exit models.” This is not true, a lot of early exiting methods can do with a single classifier heads [A-C].
>
> Thank you for pointing out these references, which have been cited in the revision. Note that none of the referred references use a single classifier for early-exit prediction.
>
> The link to [A] can’t open, but we conjecture it refers to the paper “Confident Adaptive Language Modeling”. For this paper, it uses multiple classifiers instead of a single classifier for early exit prediction. Specifically, as described in Sec. 3.2, it calculates prediction by $p(y_{t+1}| d_t^i) = softmax(W_{i}d_t^i) $, where $W_i$ represents the classifier at the $i$-th intermediate layer. It uses a single classifier only as one of three strategies when determining the threshold for exit. As noted in Sec. 3.5, the single classifier is trained through linear probing while freezing the backbone parameters. In contrast, our aligned training approach uses a single classifier for early exit prediction, jointly training it with the backbone parameters.
>
> Papers [B] and [C], on the other hand, do not involve early-exit mechanisms and therefore do not utilize multiple classifiers. Instead, these works focus on improving efficiency through whole layer pruning or fine-grained feed-forward network (FFN) pruning based on high similarity between adjacent layers. We have incorporated this discussion in the Sec. 2.1 of the revision.
>
> > Line 240: “Progressively increasing layer-wise representation similarity”, thus observations might be different in other domain[B], is there any insights why autoregressive models seems not have progressively increasing layer-wise representation similarity?
>
> We believe there is no contradiction with the observation in [B]. By difference, we conjecture reviewer refers to the results of Figure 2 in [B]. Note that Figure 2 in [B] only plots the cosine similarity **between adjacent layers**, while we measure similarity for any two layers whether they are adjacent or far apart. Indeed, our Figure 2(c,d) shows hidden layers obey larger similarity with adjacent layers, consistent with the observation in [B]. The “progressively increasing layer-wise representation similarity” refers to layer-wise similarity increases when two layers get closer, for instance, **the similarity between each hidden layer’s representation and the final layer’s representation**. We have added this discussion in footnote 3. If we misunderstood the reviewer’s comments, we would appreciate it if the reviewer could further elaborate on the comments, which we will be happy to address.
>
>
> > Missing previous literature[D] which also studies layer-wise cosine similarity, yet in the language domain.
>
> Thank you for pointing out this reference, which has been cited in the revision. To our reading, this paper studies cosine similarity between tokens from the same layer, but **does not study layer-wise cosine similarity**. The motivation of [D] is to address the over-smoothing phenomenon in transformers and graph neural networks (GNNs), where different input representations converge to nearly identical states in deeper layers, leading to high token similarity within layers. To address this issue, the work proposes hierarchical fusion methods, such as concatenation and gating mechanisms, to combine outputs from different layers and preserve representational diversity.
>
> [A] Confident Adaptive Language Modeling
>
> [B] FFN-SkipLLM: A Hidden Gem for Autoregressive Decoding with Adaptive Feed Forward Skipping
>
> [C] ShortGPT: Layers in Large Language Models are More Redundant Than You Expect
>
> [D] Revisiting Over-smoothing in BERT from the Perspective of Graph

---

> > ### Comment · Reviewer_ABue · 2024-11-27
> > **Reply to author response**
> >
> > Thank you to the authors for their response and the revisions. My concerns have been addressed, and I have updated my score to 6.

---

> > > ### Author Response · Authors · 2024-11-27
> > >
> > > Thank you for reading our rebuttal and raising score. We truly appreciate your recognition of our work.

---

### Meta-Review · Area_Chair_11V1 · 2024-12-16

**Metareview:**

This paper examines the similarity of internal representations across hidden layers for transformer models. The authors introduce a sample-wise cosine similarity metric and show that enhanced representation similarity leads to increased predicted probability and earlier saturation event, enabling multi-exit models with a single classifier.

However, according to the reviewers, the method might face weaknesses such as generalization to other tasks besides those presented in the paper, and the observation that early exist can perform reasonably well has been reported in the past.

Based on the thorough discussions and experimentations of the paper, and the statistical explanation of the observed phenomena, I would therefore suggest accepting the paper.

**Additional Comments On Reviewer Discussion:**

Reviewer ABue mentioned that the previous work also discussed multi-exit with a single classifier. However, as pointed out by the authors, the added citations are mostly on model pruning to enable early exit.

Reviewer xxse raised a solid point about the generalization of the proposed approach, which led to the added experiments by the authors. I agree with this angle and the authors have tried their best to address them during rebuttal.

Reviewer aKPD also raised a good point about model collapse across different layers in the transformer model, and the authors added a corresponding discussion in the appendix.

Reviewer s3vt was concerned about the novelty of the work. The authors replied that the novelties are mainly in the theoretical aspect of the work: discovering the effectiveness of the similarity metric, explaining the phenomena from a theoretical perspective, and proposing a training solution. Overall, I think the concern is well justified.

---

### Decision · Program_Chairs · 2025-01-22

Accept (Poster)